# Fast plasmoid-mediated reconnection in a solar flare

Xiaoli Yan [1,2,3 ✉], Zhike Xue [1,3], Chaowei Jiang [4 ✉], E. R. Priest[5], Bernhard Kliem [6], Liheng Yang[1,3], Jincheng Wang[1,3], Defang Kong[1,3], Yongliang Song[7], Xueshang Feng [2] & Zhong Liu[1,8]

Magnetic reconnection is a multi-faceted process of energy conversion in astrophysical, space and laboratory plasmas that operates at microscopic scales but has macroscopic drivers and consequences. Solar flares present a key laboratory for its study, leaving imprints of the microscopic physics in radiation spectra and allowing the macroscopic evolution to be imaged, yet a full observational characterization remains elusive. Here we combine high resolution imaging and spectral observations of a confined solar flare at multiple wavelengths with data-constrained magnetohydrodynamic modeling to study the dynamics of the flare plasma from the current sheet to the plasmoid scale. The analysis suggests that the flare resulted from the interaction of a twisted magnetic flux rope surrounding a filament with nearby magnetic loops whose feet are anchored in chromospheric fibrils. Bright cusp-shaped structures represent the region around a reconnecting separator or quasi-separator (hyperbolic flux tube). The fast reconnection, which is relevant for other astrophysical environments, revealed plasmoids in the current sheet and separatrices and associated unresolved turbulent motions.

[1] Yunnan Observatories, Chinese Academy of Sciences, Kunming, Yunnan 650216, People's Republic of China. [2] State Key Laboratory of Space Weather, Chinese Academy of Sciences, Beijing 100190, People's Republic of China. [3] Center for Astronomical Mega-Science, Chinese Academy of Sciences, 20A Datun Road, Chaoyang District, Beijing 100012, People's Republic of China. [4] Institute of Space Science and Applied Technology, Harbin Institute of Technology, Shenzhen 518055, People's Republic of China. [5] School of Mathematics and Statistics, University of St Andrews, St Andrews KY16 9SS, UK. [6] Institute of Physics and Astronomy, University of Potsdam, Potsdam 14476, Germany. [7] Key Laboratory of Solar Activity, National Astronomical Observatories, Chinese Academy of Sciences, Beijing 100012, People's Republic of China. [8] University of Chinese Academy of Sciences, Yuquan Road, Shijingshan Block, Beijing 100049, People's Republic of China. ✉email: yanxl@ynao.ac.cn; chaowei@hit.edu.cn

olar flares result from the most dramatic events of energy release by magnetic reconnection on the Sun and provide a broad range of observational detail about this fundamental process[1–3]. In the standard flare model for eruptive events[4–7], magnetic reconnection is initiated in a vertical current sheet spawned on the underside of an erupting flux rope which is associated with a coronal mass ejection (CME). Rising arcades of reconnected hot, initially cusp-shaped magnetic loops (flare loops), rooted in a pair of bright flare ribbons, are formed in the downward reconnection outflow. Confined flares result when the erupting flux is impeded by strong overlying flux or when neighboring flux systems are pressed against each other, and a reconnecting current sheet is formed at the interface[8–10]. A filament extending along the polarity inversion line of the source region often traces the activated flux. These large-scale geometries of a reconnecting current sheet, its drivers and its consequences are well established[11–20].

High-resolution imaging observations provide rich information about details of the reconnection process which result from its internal dynamics, especially the formation of plasmoids[21,22] and turbulent structures[23], and from the complexity and three-dimensional (3D) nature of the solar magnetic field[24,25]. Spectral and multi-wavelength data reveal aspects of the microscopic processes, e.g., particle acceleration[26,27], the nonthermal-thermal energy partition[28], the plasmoid instability[29,30], and bursty[31] or turbulent reconnection[32,33], in addition to flows[34]. Finally, besides hot flare plasmas, the Sun also allows the reconnection of cool chromospheric structures to be studied[24,35]. However, despite this broad range of information, comprehensive simultaneous imaging and spectral observations are still limited. In particular, while individual components of the reconnection in- and outflows have often been observed[15,18,21,34,36,37], we are aware of only two events revealing all four flows[19,35] but these have no or only limited spectral information about the involved plasma, a limitation that has prevented the mode of reconnection, i.e. Petschek-like[12], plasmoid-mediated[29], collisionless[38], turbulent[39] or other possible modes[40], from being inferred.

Here we present multi-wavelength imaging and spectral evidence for magnetic reconnection in a strong confined solar flare, traced by the cool plasma of the interacting filament and chromospheric fibrils and associated with by emerging flux. The flare is unique in that it displayed all four reconnection in- and outflows, with plasmoids in the current sheet and the separatrices. These were fully covered by spectroscopic observations, and yielded the hard X-ray images from a current sheet on the solar disk. This allows us to measure both inflows and outflows and to infer the mode of reconnection from the analysis of the physical conditions in the current sheet. We also perform a state-of-the-art, data-constrained numerical simulation of the reconnection event, which supports our interpretation of the observational data.

## Results

**High-resolution imaging of the reconnection flows.** The flare of magnitude M2.2 (of 1–8 Å flux $2.2 \times 10^{-5}$ W m$^{-2}$) occurred in Active Region NOAA (National Oceanic and Atmospheric Administration) 11967 at S13E04 (S denotes latitude; E denotes longitude) on 2014 February 2, starting at 07:17 UT, peaking at 08:20 UT, and ending at 08:29 UT. This was a large active region with a $\beta\gamma\delta$ sunspot configuration[41]. Several of its main flux components were already decaying, but there was also new flux emerging into the region. Driven by these changes, many C- and M-class flares were produced during its passage across the solar disk[42,43].

Figure 1a shows an Hα image (acquired at center wavelength of 6562.8 Å) of the X-shaped flaring region from the New Vacuum

Solar Telescope (NVST)[44] superimposed with magnetic flux contours from the Solar Dynamics Observatory (SDO)/Helioseismic and Magnetic Imager (HMI)[45] magnetogram in Fig. 1b. Using nonlinear force-free field (NLFFF) extrapolation of the photospheric vector magnetogram from SDO/HMI at 07:36 UT (see Methods, subsection NLFFF extrapolation), we obtain the relevant magnetic structures prior to the onset of the reconnection event. The yellow-colored field lines in Fig. 1b and corresponding dotted yellow path in Fig. 1a on the west (right) side of the flaring region coincide with the filament involved in the subsequent reconnection. The field lines clearly reveal the structure of a twisted flux rope[46], which is consistent with the NVST observations. The threads of the filament intertwined with one another along the filament spine (see Supplementary Fig. 1). According to the NLFFF, the flux rope has a larger cross section of ~10 Mm diameter, extending over a height range of $z \sim 5$–15 Mm at the $x$–$y$ position where the reconnection is observed (Supplementary Figs. 2 and 3). Note that the Z-axis of Supplementary Figs. 2 and 3 denotes the height perpendicular to the $x$–$y$ plane. The pink-colored field lines and corresponding path trace a set of magnetic loops on the east (left) side. Their northern part follows chromospheric fibrils but they extend to coronal heights, so are only rooted in the fibrils. These flux systems run close to each other and nearly antiparallel, suggesting the formation of a current sheet between them, in a height range $z \sim 5$–15 Mm.

An episode of magnetic reconnection, which clearly involves the filament, is displayed in Fig. 1c–h and the related Supplementary Movie 1, using Hα images from the NVST. The white arrows (the longer arrows) indicate a thread of the filament that moves into the reconnection region, reconnects across the current sheet to form a new thread in the north outflow region (blue arrows, shorter arrows), and finally moves away from the reconnection region. Since filaments are structures suspended in the corona, the reconnection must proceed at coronal heights. The cool Hα-absorbing material of the filament threads turns into emission in the coronal current sheet and cusps as a result of the heating by this local reconnection.

Figure 2 compares the imaging of the reconnection process between Hα and the extreme ultraviolet (EUV) and UV observations from the SDO/Atmospheric Imaging Assembly (AIA)[47]. The associated Supplementary Movie 2 shows an animation of these data and the additional AIA 94, 193, 211, and 335 Å channels. The white arrows in Fig. 2a point out the filament (dotted yellow line) and loop system (dotted pink line) which approached each other and began to reconnect. A bright linear structure formed between them and is continuously seen in the Hα images during 08:05–08:10 UT (blue arrows in Fig. 2f, k and Supplementary Movie 1). This is co-spatial with a nearly vertical current sheet in our data-constrained magnetohydrodynamic (MHD) modeling (see Results, subsection MHD simulation). Cusp-shaped brightened areas with newly formed threads (black arrows in Fig. 2f) extend from both ends of the linear structure. The filament is also imaged by the AIA 304 and 171 Å data (Fig. 2b, c), and the bright current sheet with cusps (yellow arrow in Fig. 2l) is seen at the same position also in all AIA wavebands, albeit with much less detail due to their lower resolution and saturation of the brightest parts (see Results, subsection Plasmoids); The bright structure indicated by the yellow arrow in Fig. 2l is not a classical Hα ribbon. It is caused by reconnection in the coronal part of the current sheet that heats the plasma in the current sheet and its attached cusps, which themselves become bright in EUV/UV and Hα, and this reconnection also provides the energy required for the other emissions.

Since the Hα images are not saturated in intensity, they reveal substantial detail of the flaring region. In particular, thread-like

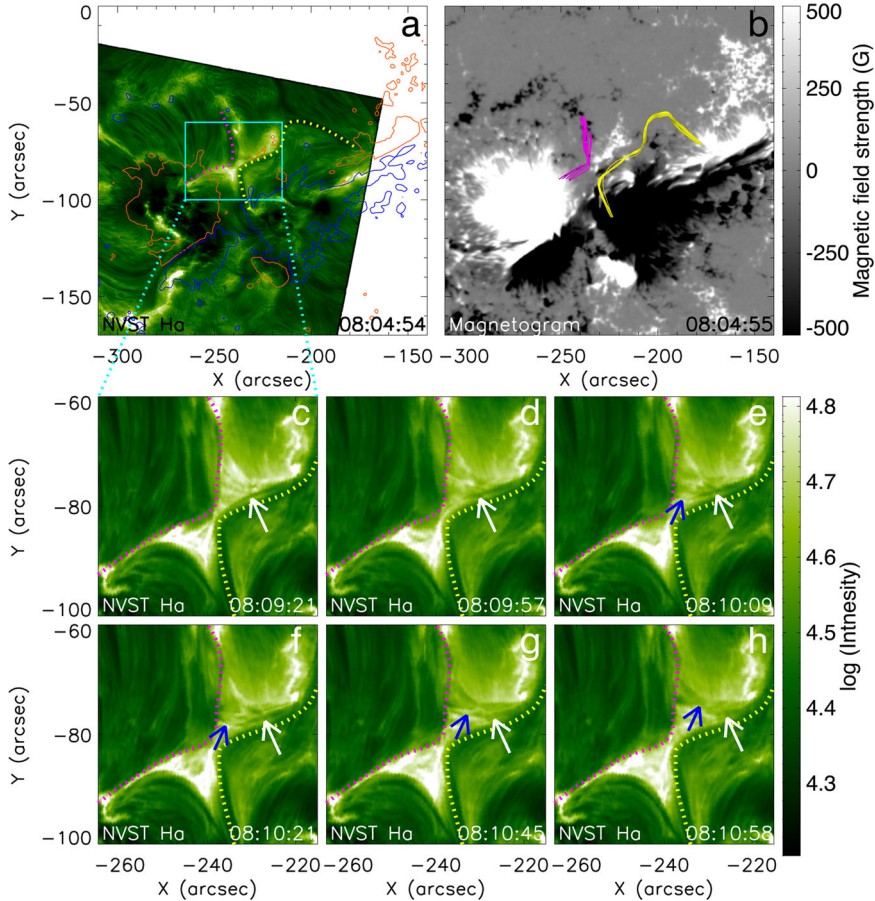

**Fig. 1 Involvement of the filament in magnetic reconnection during the confined flare, as observed by the NVST in the center of the Hα line. a** Hα image at 08:04:54 UT, superimposed by the line-of-sight magnetogram from SDO/HMI at 08:04:55 UT, with positive (red) and negative (blue) contours drawn at ±500 G. The dotted yellow and pink lines mark the filament and relevant chromospheric fibrils, respectively. **b** SDO/HMI line-of-sight magnetogram with field lines from the NLFFF extrapolation superimposed. The selected yellow lines trace the twisted flux of the filament. The pink lines show the relevant loops rooted in chromospheric fibrils. **c–h** Reconnection of a filament thread in the current sheet. White arrows indicate the thread, first in the inflow region (**c–d**), then in the outflow region (**g–h**). Blue arrows in **e–h** indicate the newly formed connection of the thread with the loops rooted in the chromospheric fibrils on the other side of the current sheet. Note that the east and west directions described in the text correspond to the left and the right sides of the Figures. Source data are provided as a Source Data file.

bright structures appear from 08:07:20 UT throughout the flare phase (up to 08:24:17 UT; also see Supplementary Movie 1). Their rapid changes from frame to frame clearly indicate structures in the coronal reconnection region, with heated filament threads being the most likely origin. During ~08:13–08:20 UT, which includes the impulsive flare phase (~08:17–08:20 UT), there is a prominent decrease of the Hα absorption in the whole area of the current sheet and cusps, suggesting they are products of heating in the separatrices or quasi-separatrix layers (QSLs) that branch off from the reconnecting current sheet and enclose the cusps[40]. This appears as a brightening very similar in shape to the EUV brightening when the display of the Hα data is artificially saturated.

Additionally, the Hα images show stationary brightenings that appear as extensions from the cusps in projection (contours in Fig. 2p–t and cyan arrows in Fig. 2q). These form where the QSLs, which extend from the current sheet in the MHD model, intersect the magnetogram plane at somewhat remote locations from the current sheet and cusps. From their essentially linear shape and stationarity and from their magnetic connection to the current sheet in the corona (detailed in Supplementary Fig. 4), we interpret these brightenings as classical chromospheric flare ribbons (see the cyan arrows in Fig. 2q), which are fed by the energy release in the coronal reconnecting current sheet. The 3D

field lines in the MHD model at $t = 42$ s show that the coronal current sheet in the range of significant reconnection ($z \sim 3$–15 Mm, cyan to red colors) is magnetically connected to the observed chromospheric flare ribbons as seen in Hα at 08:09:21 UT (see Supplementary Fig. 4). As usual, cospatial ribbons are seen in all EUV/UV channels that image plasma at coronal temperatures, and the agreement with Hα is best at 1600 Å. Overall, both Hα and the EUV/UV images show, consistently with each other, a superposition of the coronal reconnection volume (reconnecting current sheet and cusps) and chromospheric flare ribbons.

Finally, the AIA 131 Å images in the fourth column of Fig. 2 show the plasma heated to typical flare temperatures of ~10 MK. During the flare (Fig. 2d, i, n), this plasma outlines the X-shaped structure of the current sheet, cusps, and its adjacent separatrices, fully consistent with the Hα and lower-temperature EUV structure. The point at the tip of the cusps represents in 3D a separator or quasi-separator line at which two pairs of separatrices or quasi-separatrices intersect or touch. Figure 2s shows three sets of flare loops. Two sets connect the flare ribbons in the area of the points E, F, and G in Fig. 2p, suggesting they are products of heating in the QSLs. The third set is slightly westward displaced from the reconnection region and has a different origin: in addition to the long-lasting reconnection process, a partial,

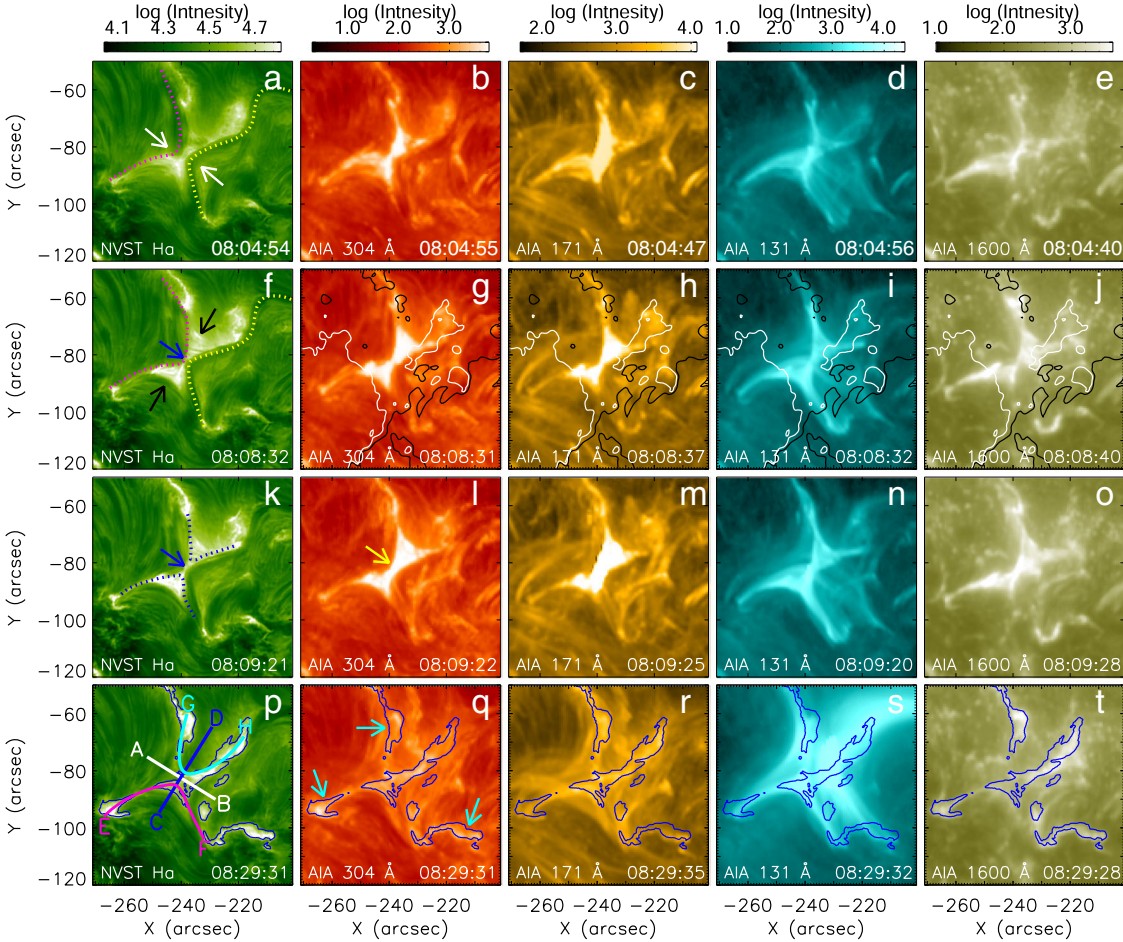

**Fig. 2 Images of the reconnection process in the center of the Hα line and in characteristic EUV lines.** The Hα images in the first column (**a**, **f**, **k**, **p**) show the reconnecting filament (dotted yellow line), magnetic loops rooted in chromospheric fibrils (dotted pink line), the current sheet region (blue arrows), and the cusp-shaped magnetic separatrices (blue dotted lines). White (black) arrows mark the reconnection inflow (outflow/cusp) regions. The lines AB, CD, EF, and GH in panel **p** are used to measure the reconnection flows. The bright structures in panel **p**, mostly flare ribbons, are marked with blue contours that are overlaid in panels **q–t** for comparison. Columns 2–5 display nearly simultaneous AIA images at 304 (**b**, **g**, **l**, **q**), 171 (**c**, **h**, **m**, **r**), 131 (**d**, **i**, **n**, **s**), and 1600 Å (**e**, **o**, **t**). Contours at ±500 G of the magnetogram at 08:08:40 UT are superimposed in panels **g–j**. The yellow arrow in panel **l** denotes the bright structure. Note that the east and west directions described in the text correspond to the left and the right sides of the figures. Source data are provided as a Source Data file.

confined eruption of the filament occurs during the impulsive flare phase. Unfortunately, the eruption is not traced by the Hα data which image only the coolest parts of the filament that do not participate in the eruption. However, the draining of filament material after the eruption is obvious in Hα, due to the strong cooling of the material (Supplementary Movie 1). The eruption can be seen in the AIA 171 and 304 Å data (Supplementary Movie 2), but is considerably masked by detector saturation, so that the onset time cannot be determined unambiguously. An onset between 08:14 UT and 08:17 UT appears likely. The Hα movie shows that the erupted part of the filament finds its final position around 08:32 UT and is then similar in shape to the third set of hot flare loops seen in the 131 Å channel. Both their southern footpoint area seen in Supplementary Movie 2 and their northern footpoint area (outside the field of view of the animation) coincide with the filament footpoints. Both branches of the filament (the remaining and the erupted but halted one), as well as the draining of the halted filament toward its southern footpoint are clearly seen in Supplementary Movies 1 and 2. It is clear also from this part of the event that Hα and the EUV (304 and 171 Å) show one and the same filament and provide consistent information about its evolution.

The partial eruption of the filament is a process separate from the reconnection studied here. This can also be seen from the GOES soft X-ray light curve in Fig. 3a and the Reuven Ramaty High-Energy Solar Spectroscopic Imager (RHESSI)[48] X-ray counts in the 3–6, 6–12, 12–25, and 25–50 keV bands in Fig. 3b, which show a short impulsive flare phase simultaneous with the partial filament eruption (~08:17–08:20 UT) superimposed on the long-lasting slow-rise phase dominated by the reconnection analyzed here. This phase reaches the GOES level M1.2.

The Hα data allow us to measure the projected reconnection in- and outflow velocities using time-slices of the intensity along the artificial slits A-B and C-D in Fig. 2p. Up to 08:14 UT, the inflows are found to be $v_i = 0.06$ km s$^{-1}$ on the east (left) side and $v_i = 0.05$ km s$^{-1}$ on the west (right) side (see the cyan lines in Fig. 3c). Here we have assumed that the inflow velocity of the loops on the east (left) side into the coronal current sheet is of the same magnitude as the velocity of the chromospheric fibrils in which the loops are rooted. The outflows are measured from $v_o = 2.0$ to 3.7 km s$^{-1}$ in southward direction, which is presumably downward, and $v_o = 3.4$ to 3.8 km s$^{-1}$ in northward direction, which is presumably upward; see the cyan lines in Fig. 3d. The resulting reconnection rate $M = v_i/v_o = 0.01$–0.03,

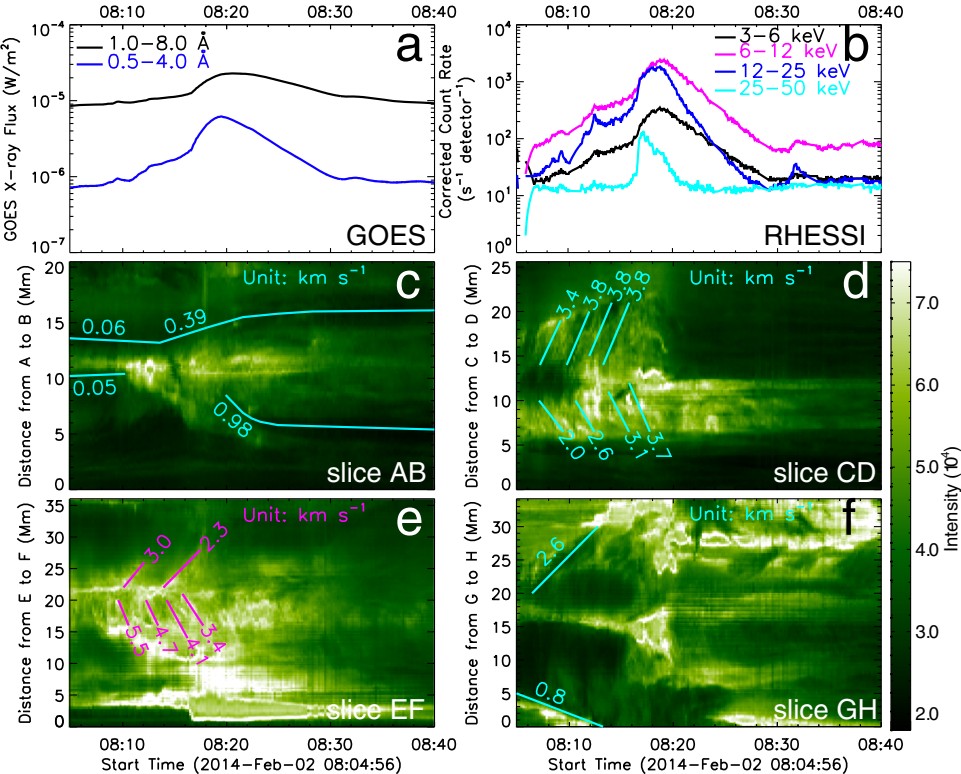

**Fig. 3 Profiles of GOES X-ray flux, inflows, and outflows during reconnection. a** The profile of GOES X-rays in 1–8 Å (black line) and 0.5–4 Å (blue line). **b** The profile of the RHESSI X-ray counts in the 3–6, 6–12, 12–25, and 25–50 keV bands. **c** Inflow (solid curve) along the A-B slit inferred from time-slices of the intensity. **d** Outflow along the C-D slit. **e** Outflow along the curved line E-F. **f** Outflow along the curved line G-H. All these lines are marked in Fig. 2p. Note that the y-axis denotes the direction information in the time–distance maps of Figure 3c-f. Note that the east and west directions of Figure 3c-f described in the text correspond to the left and right sides of the figures in Fig. 2. Source data are provided as a Source Data file.

which clearly indicates a fast reconnection process. We also measure the outflows along the cusp lines (magnetic separatrices) attached to the ends of the current sheet (see the curved lines E-F and G-H in Fig. 2p). These are found to lie in the ranges 2.3–5.5 km s$^{-1}$ (pink lines) and 0.8–2.6 km s$^{-1}$ (cyan line) for the south and north cusp lines, respectively (see Fig. 3e, f). As the reconnection continues, an unusual feature of the event is that the chromospheric fibrils and the main body of the filament began to separate from the reconnection region after 08:14 UT with velocities of 0.98 and 0.39 km s$^{-1}$ on the east (left) and west (right) sides, respectively (see Fig. 3c), whereas some of the filament threads still participated in the reconnection process. This is considered further in the Discussion. Note that the y-axis denotes the direction information in the time-slice maps of Fig. 3c, d, e, and f.

**EUV spectroscopy of reconnection flows and turbulence.** The Hinode/EUV Imaging Spectrometer (EIS)[49] observations covered essentially the whole reconnection event. Figure 4 shows the intensity (a–c), Doppler velocity (d–f), and non-thermal velocity (g–i). There is a strong red shift in the southern cusp-shaped structure, with the maximum Doppler velocity reaching about 80 km s$^{-1}$, and blue shift in the northern cusp reaching about 40 km s$^{-1}$. Although the blue shift is weaker and far less extended, it represents a definite measurement in the raster at 08:12:43 UT (Fig. 4f) because it is of uniform sign in a contiguous range of pixels (each pixel represents an independent spectral fitting) with a magnitude much larger than the noise level (estimated to be 3 km s$^{-1}$)[50]. The red and blue shifts are consistent with the NLFFF extrapolation and MHD modeling, which indicate that the north edge of the current sheet is higher than the south edge. Moreover,

the projected reconnection outflows derived from the moving fibrils are higher in the north region, suggesting a higher local Alfvén speed or motion away from the photosphere or both. However, the true peak velocity may be higher at the north side (as also indicated by the projected velocities), because the selected spectral line obviously samples only part of the upward reconnection outflow, due to a reduction of density and temperature (adiabatic cooling) in the free upward expansion of the plasma. Nevertheless, both reconnection outflows are spectroscopically detected and prove to be significantly higher near the edges of the current sheet than the outflows inferred from the Hα structures further away from the current sheet (Figs. 2p and 3d). It is also clear that the outflow is mainly downward at the southern edge of the current sheet and mainly upward at the northern edge. The implied reconnection rate, $M \sim 0.0007$ for the southern outflow, is still fast.

The cusp-shaped structures also appear in the non-thermal velocity maps. The highest non-thermal velocities were located in the current sheet, especially in the tip of the cusp-shaped structures, with values in the range ~180–300 km s$^{-1}$, which significantly exceed the projected and Doppler velocities of the reconnection outflows. This suggests that the reconnection proceeds in a highly dynamic manner including unresolved turbulent motions, with plasmoid-mediated reconnection[51,52] being the obvious candidate for the specific mode of reconnection realized in the event.

To determine the electron density in the current sheet, we used the theoretical line intensity ratios of the Fe XIV ($\lambda264.92/\lambda274.37$) density sensitive pair, which were calculated by using the CHIANTI atomic package[53,54]. Although the line $\lambda$ 274.37 is blended with Si VII $\lambda274.18$, the blend can be safely ignored due

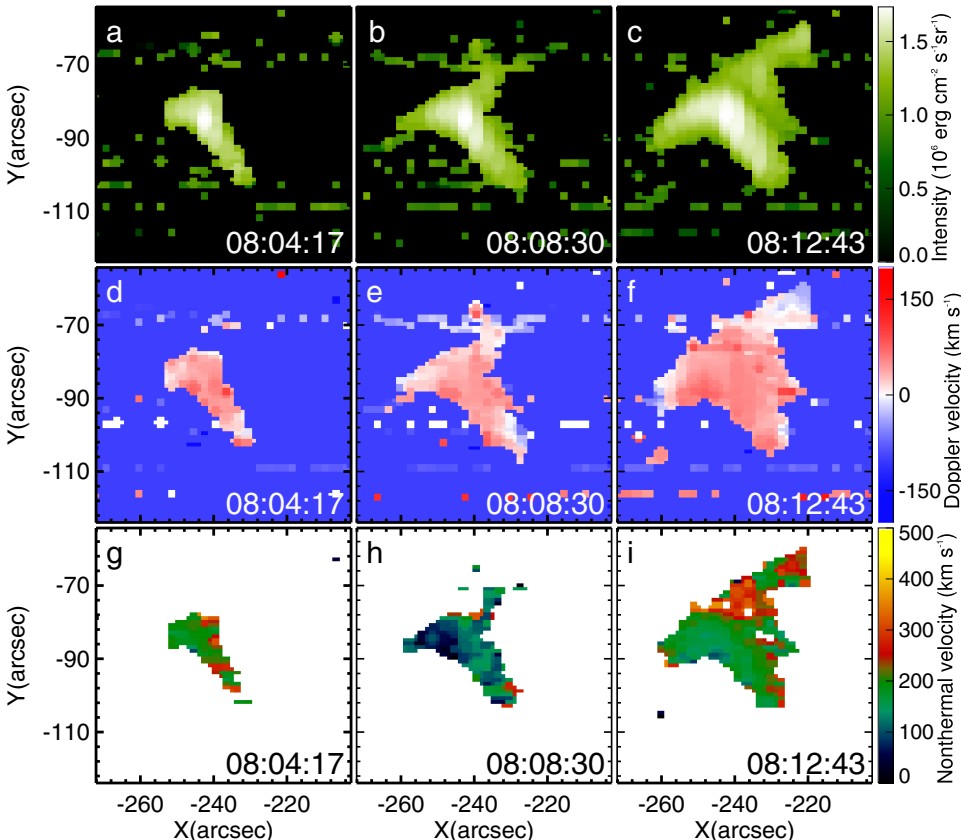

**Fig. 4 Hinode/EIS spectroscopic observation at Fe XVII 255 Å during magnetic reconnection. a–c** Evolution of the cusp-shaped structure in the intensity images. **d–f** Doppler velocity and **g–i** non-thermal velocity obtained from single Gaussian fitting. Source data are provided as a Source Data file.

to its low intensity compared with $\lambda$ 274.37 in active region conditions. The number density values in the current sheet are found to scatter in the range from $10^8$ to $10^{12}$ cm$^{-3}$, similar to typical densities in filaments[55]. The highest values are located in the southern (lower) cusp.

**Plasmoids**. During the early slow-rise phase of the flare, from 07:57 to ~08:06 UT, many small bright blobs were observed, moving from the current sheet to the legs of the cusp-shaped structures. Figure 5 and the related Supplementary Movie 3 show the highlighted fine structures of the SDO/AIA 211 Å images revealed by the unsharp masking technique (see Methods, subsection Unsharp masking technique). There are many bright blobs marked by the yellow arrows moving from the reconnection region (see Fig. 5a–f). We suggest that the blobs are plasmoids created by secondary tearing (the plasmoid instability) in the reconnecting current sheet that develops at the separator or quasi-separator: in 2D these would be magnetic islands, but in 3D with an extra magnetic component out of the plane they become tiny twisted flux tubes, as clearly shown in the MHD simulation (see Results, subsection MHD simulation). The velocities of the blobs along the reconnected field lines are from 79 km s$^{-1}$ to 208 km s$^{-1}$ (see the time-distance diagram, Fig. 5g, obtained along the cyan-dotted line in Fig. 5a). Note that the $y$-axis denotes the direction information in the time-distance maps of Fig. 5g. The phenomenon that blobs move from the reconnection region along the separatrices, as found in a 3D particle simulation of reconnection[56], is observed here. Their number, more than ten, is among the highest reported so far for a single reconnection event[21,57]. They support the interpretation of the high Fe XVII line widths as an indication of plasmoid-mediated reconnection. The increasing brightness and source width of the emissions

prevented the detection of blobs by unsharp masking at later times, but their continued formation is suggested by the continued presence of high non-thermal velocities in the EIS spectra, efficient particle acceleration revealed by the hard X-rays, and the numerical modeling below.

**Heating**. The heating and brightening resulting from the reconnection can also clearly be seen in all AIA wavebands. The Supplementary Movie 2 shows the evolution of the EUV and UV observations at eight wavelengths and of the line-of-sight magnetogram. The cusp-shaped regions of outflowing hot plasma are obvious, as is the current sheet between them. Six EUV wavelengths observed by AIA were used to obtain the differential emission measure (DEM), total EM, and temperature maps shown in Fig. 6a–f. The X-shaped geometry of the reconnection region can be seen in the whole temperature range of $T > 1$ MK, and the peak temperature in the reconnection region reaches about 10 MK.

**X-ray spectroscopy of accelerated particles**. The X-ray sources from RHESSI in Fig. 6g–i show that X-ray emission >3 keV is emitted from the beginning of the event's slow-rise phase (Fig. 6g). Higher energies are reached following the development of the flare, and the highest energy release rate into hard X-rays around 08:18 UT coincides with the steepest slope of the soft X-ray light curve (Fig. 3a), as is typically the case in flares. The emission comes mainly from the current sheet, different from eruptive flares, where the foot point sources are usually strongest[3]. Thus, the hard X-rays provide a direct diagnostic of the conditions in the current sheet. These are the maps of hard X-ray emission from a reconnecting current sheet in a confined

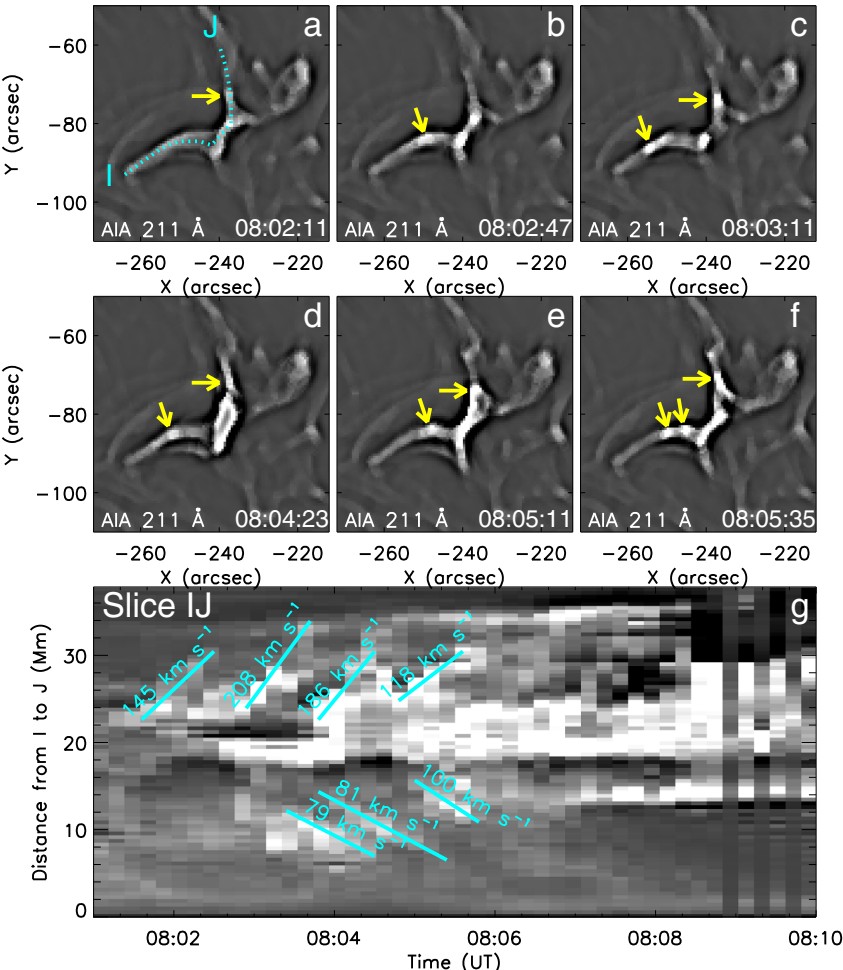

**Fig. 5 Plasmoids in the SDO/AIA 211 Å images revealed by the unsharp masking technique.** Yellow arrows mark the different bright blobs moving from the reconnection region (**a–f**). The cyan-dotted line in **a** indicates the trajectory I-J used to make the time-distance diagram of **g**. The projected velocities of the blobs lie in the range from 79 km s$^{-1}$ to 208 km s$^{-1}$. Note that the y-axis denotes the direction information in the time-distance maps of **g**. Source data are provided as a Source Data file.

flare on disk. The emission originates preferentially from the lower (southern) cusp, where the density in the sheet was found to be highest, similar to limb flares with occulted foot points[58].

Figure 7 shows the RHESSI hard X-ray photon spectra taken at about 08:12 UT and 08:19 UT. The temperatures derived from the fitted spectra are 13.1 MK (Fig. 7a) and 33.6 MK (Fig. 7b); this extends the results of the DEM analysis of the AIA EUV data which are limited to ≈11 MK. Moreover, the spectra show both thermal and non-thermal components. The emission above ~25 keV is of non-thermal nature, indicating particle acceleration in the current sheet by the fast reconnection process. Particle acceleration by reconnection is known to be particularly efficient when multiple plasmoids form, implying multiple X-lines[27,59,60]. This corroborates the deduction above that plasmoid-mediated reconnection was at work.

**MHD simulation.** To reveal the 3D structure and evolution of the reconnecting field, we have simulated the reconnection process using a data-constrained high-resolution MHD model (see Methods, subsection Data-constrained simulation). Figure 8 and the related Supplementary Movie 4 show the magnetic topology in a snapshot of the model immediately prior to the onset of reconnection, including the 3D structure of the central current sheet, sample field lines around the current sheet, and the associated separatrices or QSLs, which are revealed as regions of high

squashing degree Q[61,62]. The preferred locations for the formation of current sheets and therefore of reconnection in flares are either null points, separators or quasi-separators[40]; separators originate in null points or bald patches and represent the intersection of two separatrices, while quasi-separators (or hyperbolic flux tubes) represent the intersection of two QSLs. Maps of Q reveal separatrices or QSLs, but do not distinguish between them. QSLs are locations where the gradient in fieldline footpoint mapping is large, whereas the mapping gradient is infinite at separatrices.

At this point, the magnetic field lines from the main positive sunspot in the east (left) part of the active region connect to the diffuse negative polarity in the north (the pink lines in Fig. 8a), while the field lines from the smaller positive spot in the west connect to the main negative spot (the green lines in Fig. 8a). This is similar in structure to the NLFFF in Fig. 1, although the twist in the magnetic flux of the filament is not recovered by the extrapolation technique used to construct the initial condition. The main electric current is located between these two groups of field lines (Fig. 8b, c). Both the field lines and current sheet extend to somewhat greater heights at the northern edge, consistent with the spectroscopically inferred upward direction of the north reconnection outflow (Fig. 4f). Although the current sheet extends from the bottom of the box up to about 20 Mm, the reconnecting flux, as visualized by the field lines, extends mostly

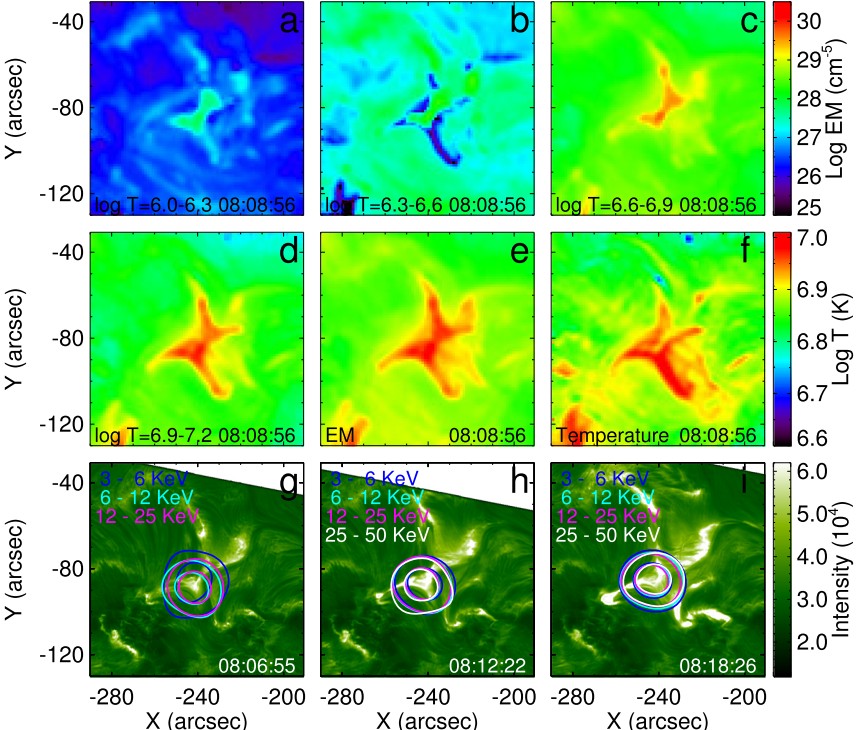

**Fig. 6 Emission measure and temperature maps and Hα images with RHESSI X-ray sources superimposed. a–d** DEM maps in four temperature ranges. **e** Total emission measure. **f** Temperature of the reconnection region derived from six AIA wavelengths. **g–i** Hα images with RHESSI hard X-ray sources superimposed. The levels of the contours are at 50% and 80% of the peak values in the 3–6 (blue), 6–12 (cyan), 12–25 (pink), and 25–50 (white) keV bands. Source data are provided as a Source Data file.

over the range $z \sim 5$–15 Mm. Outside this range, the current sheet is shorter and the reconnection flows remain weaker and less extended during the subsequent MHD evolution. In particular, no reconnection flows are observed to develop at $z \lesssim 1$ Mm. The current sheet at the chosen level of the iso-surface coincides with the QSL at the highest values of $Q$. The high-$Q$ surface outlines the separatrices or QSLs of the reconnecting flux (Fig. 8d) and aligns with the current sheet and its downward extensions to the flare ribbons, as observed in Hα (Fig. 8e) and EUV (Fig. 8f).

The current sheet and separatrices or QSLs extend to the bottom plane directly under the coronal current sheet and cusps as well as in the flare ribbons and beyond (Fig. 8b–f). However, since the field in and around the current sheet is closer to the horizontal direction than to the vertical, the current sheet is magnetically connected to the bottom plane (chromosphere on the Sun) only in the more remote traces of the separatrices/QSLs, i.e., in the flare ribbons and beyond. Supplementary Fig. 4 shows that most of the field lines from the significantly reconnecting height range in the current sheet ($z \sim 3$–15 Mm) are rooted in the observed Hα ribbons. Some are rooted in or near more distant parts of the separatrix/QSL traces, but spread out more, implying a lower density of energy deposition in the chromosphere. Since the energy transport in the corona proceeds exclusively along the field, these magnetic connections perfectly support the interpretation of the more remote brightenings (contours in Fig. 2p–t) as classical chromospheric flare ribbons energized by reconnection in the coronal current sheet.

Since this realistic, data-constrained model of the current sheet includes a guide field component in the direction of the current flow, the plasmoids seen in the current density distribution (Fig. 9b) are small, 3D flux ropes in the field line plots[56] (Figs. 9a and 10a, b). As observed in two-dimensional simulations[51,52], as well as in the observations presented above, plasmoids form dynamically and repeatedly, and are accelerated along the current

sheet. One of the plasmoids indicated by the arrows in Figs. 9b and 10a moves from near the south cusp structure to the north cusp structure. The speed of the plasmoid is approximately 300 km s$^{-1}$, which is close to the observed one. Two new plasmoids are seen to form in Fig. 9b. Between the plasmoids, the current sheet steepens to the limit allowed by the numerical resolution. The simulation clearly corresponds well to the observations from the global to the plasmoid scale. The plasmoids appeared only after adaptive mesh refinement was applied to permit the current sheet width to shrink to an aspect ratio of ~ 0.01, in agreement with the theoretical prediction for the occurrence of the plasmoid instability[63].

**Released energy.** A rough estimate of the way reconnection releases the magnetic energy required to power an M-class flare (~$10^{24}$ J) can be made as follows. We expect that part of the energy is released in the reconnecting current sheet and a further part is released by the untwisting of the filament-flux rope after it is connected to the diffuse remote polarity. Suppose first that oppositely directed magnetic field of strength $B_i$ is brought in from both sides at speed $v_i$ for a time $\tau$ towards a reconnection region of surface area $L_0^2$, where the current sheet and attached slow-mode shock waves convert the energy. The magnetic energy is brought in from both sides from a volume $2L_0^2 L_i$, where $L_i = v_i \tau$, and so the total energy calculated is

$$W = \frac{B_i^2}{2\mu_0} L_0^2 v_i \tau. \quad (1)$$

Evaluating this for a field strength of ~500 G (see Supplementary Fig. 2), which is exceeded in a cross section of the filament-flux rope of diameter >10 Mm in the NLFFF, and $L_0 = 15''$ (10.9 Mm), $v_i = 2$ km s$^{-1}$, and $\tau = 70$ min as obtained from the Hα data, the

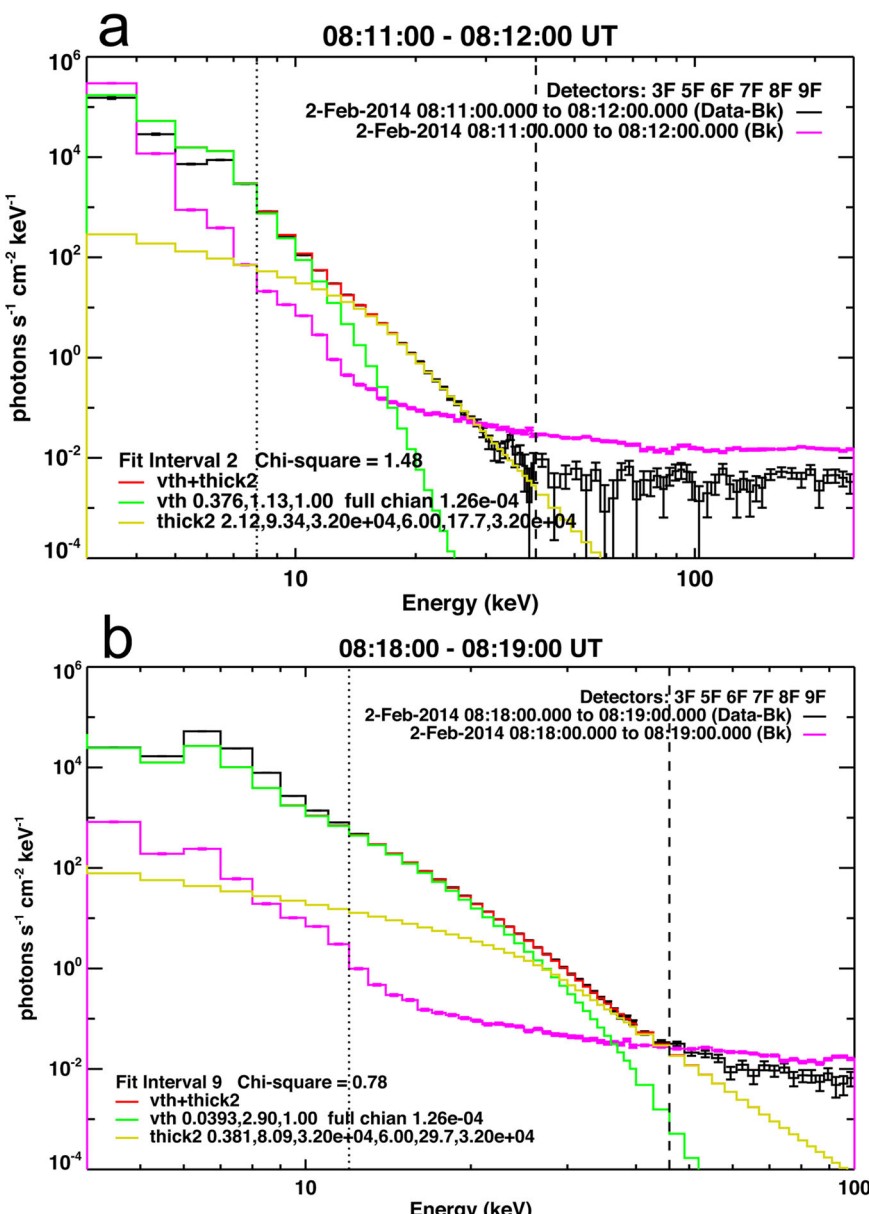

**Fig. 7 RHESSI photon spectra accumulated around 08:12 UT and 08:19 UT during the quoted 1-min intervals on 2014 February 2. a** 08:12–08:12 UT. **b** 08:18–08:19 UT. The spectra are fitted by combining the vth and thick2 functions (see Methods, subsection RHESSI spectrum). The black line shows the photon spectrum with the background subtracted. The pink line shows the background. The green and yellow lines display the fitting with the single-temperature thermal model and the thick-target nonthermal model, respectively. vth is used to fit the thermal portion of the spectra, generally from 4 keV to between 20 and 25 keV depending on the attenuator state. thick2 is used to fit Thick-Target Bremsstrahlung X-ray/gamma-ray spectrum from an isotropic electron distribution. Bk indicates the background photon flux. Data-Bk indicates the observational photon flux with the background flux subtracted. Detectors indicate the detectors that used to obtain the photon flux. The error is the square root of the counts used in the fitting. When combining time bins, these errors are averaged. Source data are provided as a Source Data file.

energy is estimated to be ~$10^{24}$ J. This corresponds well to the energy release of ~$10^{24}$ J ($9.6 \times 10^{30}$ erg) in the simulation and agrees with the canonical picture that the flare is powered by magnetic reconnection.

Additional magnetic energy may be liberated by the untwisting of the flux rope which holds the filament. The energy associated with the twist component of the flux rope field, $B_\phi = (R/a)B_{0\phi}$, can be written as

$$W_{FR} = \frac{B_{0\phi}{}^2}{2\mu_0}\pi a^2 L_R, \quad (2)$$

where $R$, $a$ and $L_R$ are the rope's major and minor radii and length. Suppose the twist on the edge of the flux rope is $\Phi$, then $\Phi = (LB_{0\phi})/(aB_z)$ and $B_{0\phi}$ is $a\Phi B_z/L_R$. In the present case, we have $L_R = 100''$ from the H$\alpha$ data, and the NLFFF extrapolation indicates that the twist peaks near the center of the flux rope, with an average value of $\Phi \sim 5\pi$ in a cross section of radius $a \approx 2.5''$ and negligible twist in the outer parts of the rope; also $B_z \sim 500$ G (see Supplementary Fig. 3). This yields a value of $W_{FR} \sim 10^{23}$ J, significantly smaller than the energy released in the reconnecting current sheet.

In summary, the presented high-resolution imaging and spectral observations demonstrate that fast magnetic

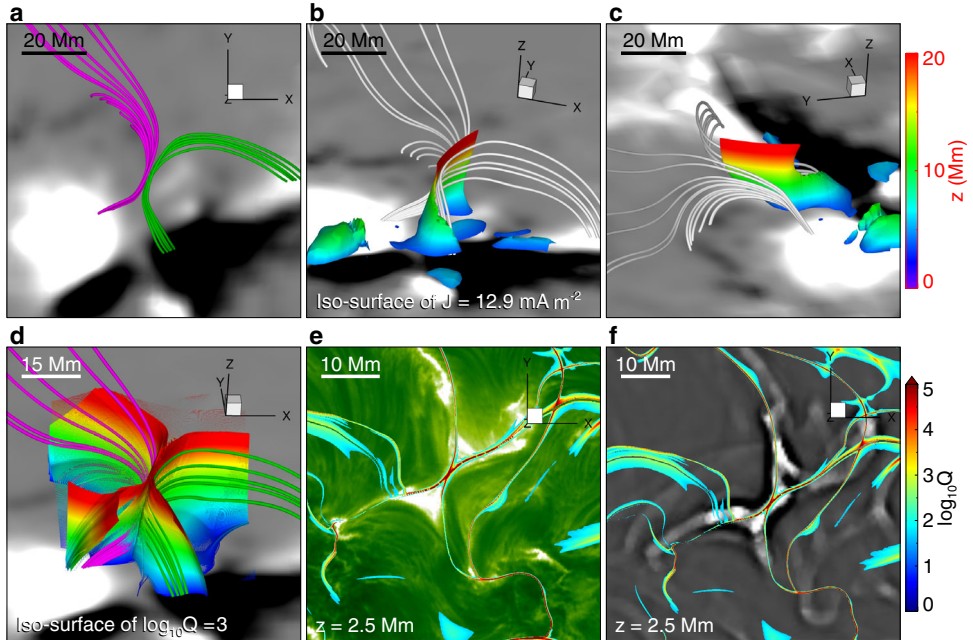

**Fig. 8 Pre-reconnection structure of the magnetic field lines, current sheet, and separatrices or QSLs, as reproduced by the data-constrained MHD model, and their relation with the observed structures. a** Selected magnetic field lines showing the magnetic structure at the side of the filament (green) and the magnetic loops rooted in the chromospheric fibrils (pink) overlaid on the longitudinal magnetogram at 08:00 UT observed by SDO/HMI. Note that the background image of panels b, c, and d is the same as panel **a**. **b**, **c** Two views of the same field lines and the current sheet between them, displayed as an iso-surface of the current density color coded with the height information. **d** The X-type structure of the separatrices or QSLs, displayed as iso-surface $\log_{10}Q = 3$ of the squashing degree $Q$, also color coded with height. **e** Horizontal map of the separatrices or QSLs (here $\log_{10}Q > 2.5$) at height $z = 2.5$ Mm superimposed on the H$\alpha$ image at 08:09:21 UT. **f** Same separatrix or QSL map superimposed on an enhanced 211 Å image at 08:05:35 UT showing the plasmoids. Note that the east and west directions described in the text correspond to the left and the right sides of the Figures. Source data are provided as a Source Data file.

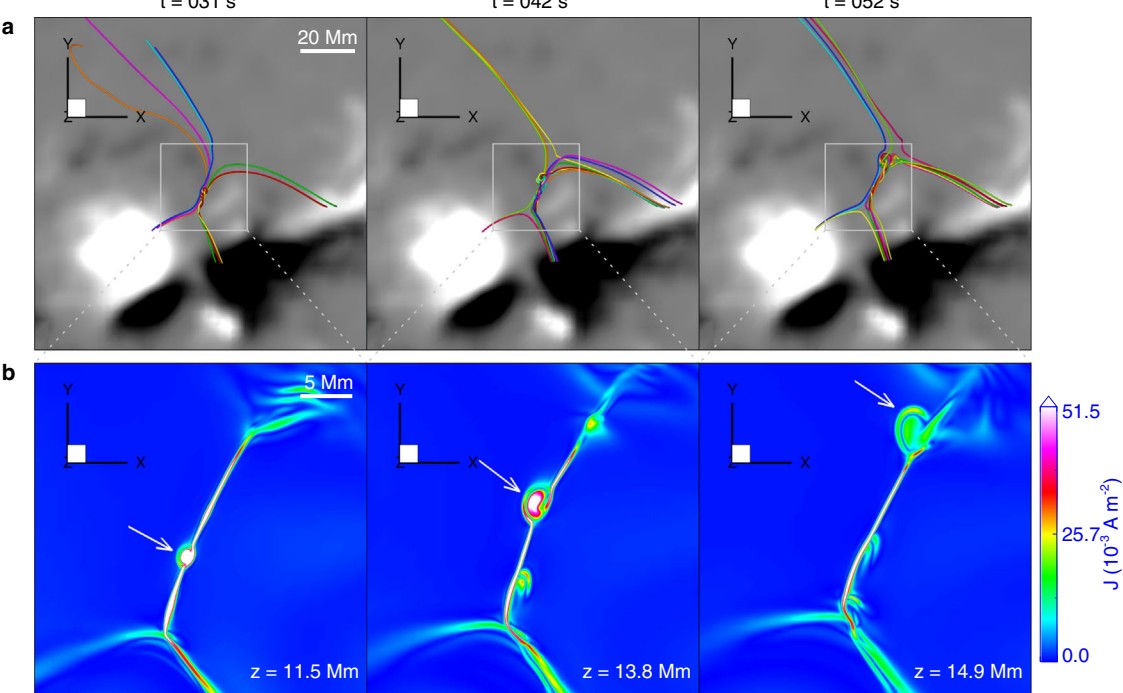

**Fig. 9 Evolution of magnetic field, current sheet, and plasmoids (mini flux ropes) during magnetic reconnection in the data-constrained MHD simulation. a** Evolution of the overall magnetic structure involved in the modeled reconnection event. Note that the field lines are randomly colored from column to column, and thus the same color does not indicate the same field line. The background image is the longitudinal magnetic field at 08:00 UT observed by SDO/HMI. **b** Distribution of electric current density at three selected heights and times in the current sheet. The heights are chosen at which the mini flux rope is sliced at its middle, i.e., the plasmoid is seen most clearly. The white arrows indicate the biggest plasmoid.

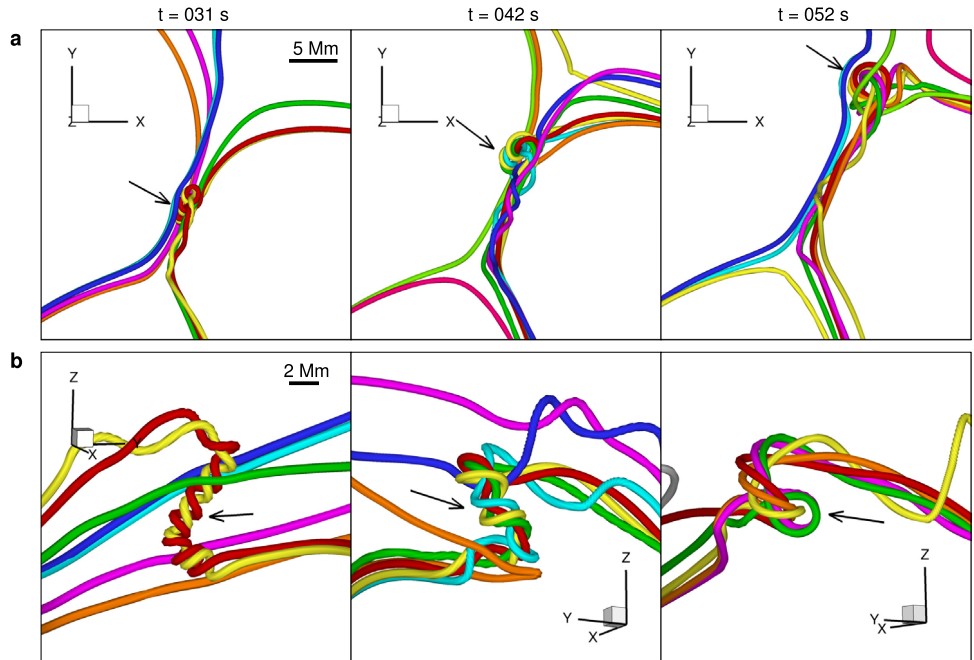

**Fig. 10 Magnetic structure of the current sheet. a** The corresponding magnetic structure of the current sheet. The black arrows indicate the magnetic structure of the biggest plasmoid. **b** 3D magnetic structure of the biggest plasmoid in a side view at three selected times. The arrows mark the heights given in panel **b** of Fig. 9. Note that the magnetic field lines are shown by the thick colored lines, and the colors are used for a better visualization of the different lines.

reconnection of the coronal field, part of which supports a filament at chromospheric temperatures, released the magnetic energy which powered the solar flare. Plasmoid-mediated reconnection is strongly indicated by direct imaging and nonthermal EUV and hard X-ray spectra. This is further supported by a 3D data-constrained MHD simulation which reproduces the observed change in connectivity, agrees with estimates of the energy release, and reveals the appearance of twisted plasmoids (mini flux ropes) at the theoretically expected threshold.

## Discussion

In order to understand why reconnection occurred so many times in this place, the evolution of the magnetic and flow fields in the photosphere was investigated using HMI vector magnetograms. The vector magnetogram is shown in Supplementary Fig. 5a–c, where the blue arrows represent the horizontal magnetic field. The corresponding flow field was calculated from the radial magnetic field using the DAVE algorithm[64] (see Methods, sub-section Velocity field calculation) (see Supplementary Fig. 5d–f and the related Supplementary Movie 5). This reveals a flow toward the magnetic reconnection site from the west (right) side, which pushes together oppositely directed flux in the lower corona (as found in the MHD simulation). This persistent flow in the photosphere is likely to be the main reason why so many flares (three M-class flares and several C-class flares of similar morphology in the first half of 2014 February 2) occurred in the same place. Additional distortions of the coronal field configuration, also likely to support the onset of reconnection, are produced by the emergence of flux very near the reconnection site (yellow ellipses in Supplementary Fig. 5b, c) and subsequent motion toward the reconnection site. The flows at the east (left) side are more variable, with directions along and away from the current sheet alternating. The latter is the most likely explanation of the separation of the reconnecting flux after ~08:14 UT in the present

event, as indicated by the retraction of the chromospheric fibrils at the east (left) side of the current sheet (Fig. 3c), as well as for the termination of the other homologous events on the same day.

For the specific event studied here, the coronal field was also strongly perturbed by the partial, confined eruption of the filament during the peak phase of the flare. The confinement has likely led to the termination of the reconnection inflows at the west side of the current sheet (Fig. 3c).

A neighboring eruption may have had an additional influence. The M-class flare studied here was preceded by a weaker, C-class flare above another polarity inversion line in the complex active region, lying to the east of the main positive sunspot, at $(x, y) \approx (-340, -50)$. The C-class flare was caused by the rise of flux in the corona above the filament located at that inversion line. Temporarily, the associated large-scale change of the coronal field, due to the expansion of that flux, may have played an additional role in pushing the magnetic loops rooted in the chromospheric fibrils toward the filament.

## Methods

**Observational data**. Images in the center of the Hα line at 6562.8 Å obtained at the 1 m New Vacuum Solar Telescope (NVST) at Fuxian Lake[44], are used in this study. The spatial resolution of Hα line center images is a pixel size of 0.″163 and the time cadence is 12 s. The Level 1 data are obtained by using the raw data (Level 0) subtracting the dark current and flat field and then we used the speckle masking method[65] to reconstruct these Level 1 data to get Level 1+ data.

Full-disk UV and EUV images observed by SDO/AIA[47] are also used to clarify the process of magnetic reconnection. These images have a 12 s cadence and a spatial resolution of 0.″6 per pixel. Vector magnetograms from the Space Weather HMI Active Region Patch (SHARP) series observed by the Helioseismic and Magnetic Imager (HMI)[45,66,67] on board SDO are used to show the evolution of the magnetic field in the photosphere, to infer the photospheric velocity field, and to perform a data-constrained simulation of the event. The vector magnetograms have a pixel scale of about 0.″5 and a cadence of 12 min. They are derived by using the Very Fast Inversion of the Stokes Vector algorithm[68]. We used the minimum energy method[69,70] to resolve the 180 degree azimuthal ambiguity of the vector magnetograms. The images are remapped using a Lambert (cylindrical equal area) projection centered on the midpoint of the AR, which is tracked at the Carrington rotation rate[71].

The spectral data observed by the EUV Imaging Spectrometer (EIS) on board the Hinode mission were used to derive the Doppler velocity and non-thermal velocity in the reconnection region. Hinode/EIS[49] is a scanning slit spectrometer that uses two EUV wave bands. One is 170–210 Å and the other is 250–290 Å. Its spectral resolution is about 0.0223 Å pixel$^{-1}$. The accuracy of the obtained velocities can reach a few km s$^{-1}$. Fortunately, the EIS observation covered the whole duration of the reconnection event and nearly its whole area. Therefore, it is well-suited for conducting a study of magnetic reconnection. We use the line Fe XVII 255 Å (log $T = 6.6$). The field of view (FOV) is 161″ in the raster direction and 152″ in the slit direction. The spectral profiles are fitted with single Gaussian functions. The reference wavelength used to determine the Doppler and non-thermal velocities is taken from the average of each raster. The sparse raster used the 2″ slit with a step size of 2.″995 and an exposure time of 4–6 s.

Data from RHESSI[48] are also employed to study the soft and hard X-ray sources during the M-class flare. The energy ranges of 3–6, 6–12, 12–25, and 25–50 keV were selected to observe the evolution of these sources during the magnetic reconnection. We used the clean imaging algorithm to construct the X-ray images and the accumulation time is about 30 s.

**Data-constrained simulation**. We solve the fully 3D and time-dependent equations with bottom boundary conditions driven continuously[72,73] by the evolving photospheric vector magnetic field taken by SDO/HMI (see Observational data). The initial magnetic configuration is obtained from the SDO/HMI vector magnetogram at 08:00 UT using an MHD relaxation technique[73], and the background coronal atmosphere is initialized by a isothermal plasma ($T = 10^6$ K, corresponding to a sound speed of $c_S = 128$ km s$^{-1}$) in a hydrostatic state. In order to mimic the coronal low-$\beta$ and highly tenuous conditions, we use a plasma density to ensure that the smallest value of plasma $\beta$ is $2 \times 10^{-3}$ (and thus the largest Alfvén speed is 4 Mm s$^{-1}$). The plasma thermodynamics are simplified to be adiabatic. We used the magnetic field of the photosphere as the bottom boundary of this model, which is assumed to be a reasonable approximation of the coronal base. To obtain the detailed structure in the current sheet, an adaptive refinement mesh is used with highest resolution of 1/16 arcsecond (i.e., 45 km) to capture the fine structures in the reconnecting current sheet.

**Unsharp masking technique**. The unsharp masking technique[22] is applied to SDO/AIA 211 Å images to highlight fine structures like blobs. First, a background image by smoothing the original image is obtained as an unsharp masked image. Second, the enhanced image is the residual of the original image subtracting the background one. A smoothing window of $7 \times 7$ pixels is adopted for optimal effects.

**NLFFF extrapolation**. The three-dimensional magnetic structures of the filament and the magnetic loops are reconstructed by using an optimization algorithm that is developed by Wiegelman[74]. Before the NLFFF extrapolation, we modified the bottom vector magnetic field data observed by SDO/HMI by using a preprocessing procedure[75] to remove most of the net force and torque, which may lead to an inconsistency between the photospheric magnetic field and the force-free assumption in the NLFFF models. The software Paraview is used to show the visualization of the 3D magnetic field.

**RHESSI spectrum**. The Object Spectral Executive (OSPEX)[76] software is used to produce the RHESSI hard X-ray photon spectra with the background subtracted. The spectra were fitted by using the combination of the single-temperature thermal bremsstrahlung radiation function (vth) and the thick-target non-thermal bremsstrahlung with an isotropic pitch-angle distribution (thick2). The detailed explanations of the fitting parameters can be seen from https://sprg.ssl.berkeley.edu/~tohban/wiki/index.php/Vth_-_Variable_Thermal and https://sprg.ssl.berkeley.edu/~tohban/wiki/index.php/Thick2_-_Thick_Target_Bremsstrahlung_Version_2.

**Velocity field calculation**. The photospheric flow fields were derived by using the method of the differential affine velocity estimator (DAVE)[64,77] based on the data of the HMI vector magnetograms. In the DAVE method, the advection equation and a differential feature tracking technique were combined together to detect flow fields. The window size is set as 19 pixels, which is the best choice for structure information and spatial resolution.

## Data availability

The NVST data can be accessed at http://fso.ynao.ac.cn/dataarchive_ql.aspx. Photospheric magnetograms are provided by the HMI on SDO. All SDO data are publicly available at http://jsoc.stanford.edu/ajax/lookdata.html and at http://helioviewer.org/. The Hinode/EIS data are publicly available at http://solarb.mssl.ucl.ac.uk/SolarB/SearchArchive.jsp. The RHESSI data are publicly available at https://hesperia.gsfc.nasa.gov/hessidata/. Source data for all the Figures except for Figs. 8, 9 and 10 due to large size of the 3D MHD simulation data are provided with the paper. The MHD simulation datasets generated during and/or analyzed during the current study are available from the corresponding author upon request. Source data are provided with this paper.

## Code availability

The method of obtaining the GOES soft X-ray flux is publicly available at https://hesperia.gsfc.nasa.gov/rhessidatacenter/complementary_data/goes.html. The Solar Software package is available at http://www.lmsal.com/solarsoft/. The DAVE code is available at https://ccmc.gsfc.nasa.gov/lwsrepository/index.php. The detailed descriptions of using OSPEX and NLFFF extrapolation are available at https://hesperia.gsfc.nasa.gov/ssw/packages/spex/doc/ospex_explanation.htm and http://sprg.ssl.berkeley.edu/jimm/fff/optimization_fff.html. MHD simulation code used in this study is available upon request.

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

## Acknowledgements

We would like to thank the NVST, SDO/AIA, SDO/HMI, GOES, RHESSI, and Hinode teams for high-cadence data support. This work is sponsored by the National Science Foundation of China (NSFC) under the grant numbers (11873087, 11973084, 11803085, 2019FD085, 12003064, 11633008, 11763004, U1831210, 11803002), by the Yunnan Key Science Foundation of China under numbers (2018FA001, 2018FB007) and Yunnan Science Foundation for Distinguished Young Scholars No. 202001AV070004, by Project Supported by the Specialized Research Fund for State Key Laboratories, by Key Research and Development Project of Yunnan Province under number 202003AD150019, and by the joint NSFC/DFG Grant 41761134088/KL817.8-1. C.W.J. acknowledges support by National Natural Science Foundation of China (41822404, 42174200), the Fundamental Research Funds for the Central Universities (grant No. HIT.BRETIV.201901), and Shenzhen Technology Project under grant number JCYJ20190806142609035. B.K. acknowledges support by the DFG. The computing resources for the data-constrained MHD simulation was provided by National Supercomputer Center in Tianjin, and the calculations were performed on TianHe 1 A.

## Author contributions

X.L.Y. developed the ideas, performed analysis of the data, wrote the draft manuscript, and led the discussion of the manuscript. Z.K.X., L.H.Y., J.C.W., D.F.K. and Y.L.S. performed analysis of the data. C.W.J. carried out the MHD simulation. E.R.P. and B.K. contributed to the interpretation of the data and to the discussion. X.S.F. and Z.L. contributed to the discussion. All authors reviewed the manuscript.

## Competing interests

The authors declare no competing interests.
