## [Peer Review File · Nature Communications]

REVIEWER COMMENTS

Reviewer #1 (Remarks to the Author):

This is a very interesting paper reporting various fundamental features of magnetic reconnection in a solar flare, such as all four reconnection flows, plasmoids not only in the current sheet but also in the separatrices, and the first hard X-ray images from a current sheet on the solar disk. Hence I am very positive for publication of this paper in Nature Communication Journal.

However, I have several questions and comments as listed below, so I would recommend the authors to revise the paper, considering these questions and comments.

1) p 5 The authors wrote that Fig 4 and Movie 2 show some bright blobs moved from the tip of the cusp to downward and further wrote:

“To the best of our knowledge, the phenomenon that blobs move from the reconnection region along the separatrices, as found in a 3D particle simulation of reconnection[52], is observed here for the first time. “

This seems very interesting. However, I quickly checked the reference Dawton et al. (2011) and had a feeling that it is not easy to understand the 3D structure of magnetic field lines associated with the blobs. If these observed blobs are simply dense structure without magnetic field lines, it would be easy to understand them. However, the authors wrote these are plasmoids. Are they helically twisted flux rope (as usually considered for flare current sheets in 3D) ? Please answer this question. If these are magnetic structure, would you please show 3D field line configuration of the blobs in supplementary page ?

2) Movie 4 is a very misleading movie. There are many pixels showing unphysical temperature. I recommend to revise this movie or remove it from the paper.

3) In p6, the authors wrote “The (hard X-ray) emission comes mainly from the current sheet, different from eruptive flares, where the foot point sources are usually strongest.” Is this true ? Would you give the reference ?

4) In P7, the authors wrote

"Figure 7 shows the magnetic structure of the filament, the magnetic loops,,,,,"

However, it is not easy to understand this figure. Please add the information of the height in (d) and (e).

5) In relation to 4), I have a fundamental question : Why H alpha images show beautiful cusp or separatrix (X type) structure in spite of the fact that H alpha show chromospheric structure ? Figure 7 seems to show that the X-type magnetic field configuration is in the corona and high above chromosphere so it seems difficult to explain H alpha image morphology with this magnetic field configuration.

6) Fig 8 is not easy to understand. What is the height of plasmoids ? Please enlarge the part of the plasmoids, and show magnetic field configuration around the plasmoids, especially 3D connectivity of each field line, much more clearly.

7) In p 8 , the author wrote $B \sim 500G$. Is this true ? It seems a bit strong for the average field strength around the coronal current sheet. If correct, would you show us evidence based on MHD modeling ?

I would be happy if the authors would reply these questions and comments, and revise the paper so that the reader would understand this interesting paper more easily.

Reviewer #2 (Remarks to the Author):

Multi-scale Observation and Data-driven Modeling of Magnetic Reconnection in Flaring Chromospheric Plasmas on the Sun

X. Yan et al.

The authors present picture book images of a solar flare. The images are well presented and the text is well written. The high-quality H alpha pictures seem to show a Petschek type reconnection. The images show four flows along magnetic separatrices. The measured velocities match the expectations.

To good to be true? Some doubts creep in when becoming aware that flares release most energy in non-thermal particles which heat the flare plasma to millions of degrees. Yet the authors put most weight on H alpha observations, tracing cool plasma. The question needs to be addressed: How relevant is H alpha? H alpha and EUV should be compared.

The second question concerns the tracing of the magnetic field. Fibrils do not always trace the magnetic field (e.g. Asensio Ramos+ 2017). Figure 7 may suggest the answer, but needs to be discussed.

The blue-shift claimed in Fig. 3 seems to be marginal. Some more evidence needs to be presented.

Finally, the reader would like not only to be presented a well-fitting case, but to know which of the many other observations of this flare remain unexplained.

New in this paper are the special form of the flare, showing an X-shape and the high quality of the data. The paper confirms a well-known flare mechanism.

Reviewer #3 (Remarks to the Author):

The authors of this paper want to show directly magnetic reconnection observed in Halpha with blob formation in AIA 211 A during the reconnection. They have very good observations and after making a nlfff extrapolation they have done a very fine analysis.

I think that it is an interesting paper, even reconnection should occur on a very lower spatial scale than the observations may allow. With these observations, they interpret the plasmoids as being a good signature of reconnection.

The NVST movie is beautiful but unfortunately the Figure 1 does not give the same impression. It is very difficult to visualize with these snapshots what is important.

In the Figure 1 the filament on the right and the chromospheric fibrils (whirls around the sunspot). These structures should be indicated in the Figure. In the movie we observe well these structures. They are very stable and not perturbed at all by the reconnection. Therefore it is difficult to agree with the sentence of the abstract: « The flare resulted from the interaction of a twisted filament and chromospheric fibrils ». In Halpha we probably see the bright flare ribbons at the QSL footprints in the low atmosphere, the filament being unchanged. The ribbons are blocked by the flux rope of the filament on one side and by the whirls on the other side.

Looking at the same time at the movies of AIA in the different wavelengths, we have a different interpretation of this reconnection and flare than what is proposed in the paper. In fact we visualize two arcades which are reconnected in the corona with two opposite cusps. The footprints of these arcades are anchored in the QSLs between the filament and the chromospheric fibrils. The interaction with the filament and the whirls are not really visible.

The movies of AIA are not shown in a Figure and not discussed in the paper.

The last reconnected 131 A and 94 A loops are different from what is seen in Halpha at 08:29 UT in Figure 1.

In Figure 7 it is difficult to see exactly where should be the footprints of these reconnected arcades. May be we see two triangles at $z=0$ and they join at $z=2.9$ Mm. In d we see well the south arcade with the iso-surface. In e with see the arcade around the filament but it does not bring information. Vertical cuts should be done to show the two cusps.

Figure 8 is what happens at the reconnection high in the corona. It is very well done.

Definitively the photospheric motions favors the reconnection of these two systems of arcades: a very good case-study of reconnection.

In conclusion, it is a very good paper but the interpretation of the movies have to be revised or explained. A new figure with the evolution of AIA 94 A and 131 A is suitable.

The quality of Figure 1 should be improved. The flare ribbons in Halpha and the bright loops visible in 131 A should be discussed.

We thank the referees for their careful and insightful comments which have greatly helped us improving the manuscript. The present revision addresses all comments in a constructive way, as indicated in the detailed responses below. Most importantly, we have addressed the concern expressed by all three referees that it was not sufficiently clear whether not only the EUV images but also the H-alpha images map the actually reconnecting structures in the corona. To this end, first, Fig. 1 was updated to focus on a critical phase in the event whose H-alpha data unambiguously demonstrate the reconnection of a filament thread. Second, Fig. 2 and the associated text explicitly compare the H-alpha and EUV images of the event. A further major improvement consists in the more detailed presentation of the data-driven MHD model, especially with regard to the QSLs and the plasmoids/mini flux ropes. All changes are set in bold face; deleted text is marked as [...].

We have also double-checked and updated the estimate for the energy release when the filament untwists. Here we have used the twist map for the flux rope in our NLFFF (new Suppl. Fig. 2) to determine where the flux rope is most highly twisted, namely, in the rope's core, with a radius of 2.5 arcsec, which is now used to evaluate Eq. (2). As before, this is only a minor part of the energy release.

Below we respond to all comments in greater detail as possible in the manuscript text, due to the word count limit for the text. Since partly similar concerns were expressed, our responses contain some repetition.

Reviewer 1

1) p5 The authors wrote that Fig4 and Movie 2 show some bright blobs moved from the tip of the cusp to downward and further wrote:

“To the best of our knowledge, the phenomenon that blobs move from the reconnection region along the separatrices, as found in a 3D particle simulation of reconnection[52], is observed here for the first time. “ This seems very interesting. However, I quickly checked the reference Dawton et al. (2011) and had a feeling that it is not easy to understand the 3D structure of magnetic field lines associated with the blobs. If these observed blobs are simply dense structure without magnetic field lines, it would be easy to understand them. However, the authors wrote these are plasmoids. Are they helically twisted flux rope (as usually considered for flare current sheets in 3D) ? Please answer this question. If these are magnetic structure, would you please show 3D field line configuration of the blobs in supplementary page?

Reply: Yes, indeed the observed plasma blobs are highly interesting. One possibility is that they are simply thermal structures moving along the current sheet and then along the separatrix field lines. We suggest instead that they are plasmoids created by secondary tearing (plasmoid instability) in the reconnection region: in 2D these would be magnetic

islands, but in 3D with an extra magnetic component out of the plane they become tiny twisted flux tubes, as now more clearly shown in the strongly updated Fig. 8 (now Fig 9). The link to plasmoids (instead of simple density enhancements) can be made based on the facts (1) that the occurrence of the plasmoid instability is to be expected when the current sheet is long enough, i.e. when the aspect ratio exceeds about 30 [61], and (2) that no other process is known that would produce localized non-magnetic density enhancements in a $\beta < 1$ current sheet. Plasmoids formed by the plasmoid instability do have an enhanced density (e.g., Fig. 7 in [31]). Therefore, we consider the interpretation of the blobs seen in the EUV images (Fig. 5) as magnetic plasmoids to be the only viable interpretation at present. We have extended the argumentation in the manuscript text accordingly (see Results; Plasmoids on page 8).

Daughton+2011 [53] describe the formation and interaction of secondary flux ropes at the separatrices, whose interaction eventually leads to turbulent reconnection. In Fig. 3a they show a cross section of “four rope structures” which have evolved by coalescence from six original “filaments”, unfortunately, without including field lines. Some selected field lines are included at the later stage in Fig. 4; these indicate weakly twisted flux tubes and are marked as “secondary flux ropes.” Since their system already approaches turbulent reconnection at this stage, the magnetic structure is already highly complex. Nevertheless, the authors still identify flux ropes, which provides a correspondence with the plasmoids/small flux ropes in our MHD simulation (Fig. 9).

2) Movie 4 is a very misleading movie. There are many pixels showing unphysical temperature. I recommend to revise this movie or remove it from the paper.

Reply: Movie 4 is now removed.

3) In p6, the authors wrote

“The (hard X-ray) emission comes mainly from the current sheet, different from eruptive flares, where the foot point sources are usually strongest. “Is this true? Would you give the reference?”

Reply: Benz 2017 [3] writes in Sect. 2.5 “Standard flare scenario”, Item 2: “Hard X-rays (>25 keV) often originate from sources at the footpoints of the loop emitting soft X-rays.” This reference is now included next to the statement on page 9.

4) In P7, the authors wrote “Figure 7 shows the magnetic structure of the filament, the magnetic loops,,,” However, it is not easy to understand this figure. Please add the information of the height in (d) and (e).

Reply: Fig. 7 is thoroughly updated as Fig. 8, now including the height information. The figure is also extended to show the current sheet and,

more clearly, the separatrices (isosurface $\log Q=3$). A corresponding animation (Supplementary Movie 4) presents 360 deg perspective views on these structures, as well as horizontal cuts through the current sheet and separatrices in the range $z=0-16$ Mm. The current sheet extends throughout and even somewhat above this range, while the reconnecting flux, as visualized by the field lines, extends mostly in the range $z \sim 5-15$ Mm. This information is now included in the text or figure caption.

5) In relation to 4), I have a fundamental question: Why H alpha images show beautiful cusp or separatrix (X type) structure in spite of the fact that H alpha show chromospheric structure? Figure 7 seems to show that the X-type magnetic field configuration is in the corona and high above chromosphere so it seems difficult to explain H alpha image morphology with this magnetic field configuration.

Reply: Although H-alpha shows plasma of chromospheric temperatures, this plasma is not necessarily confined to the height range of the chromosphere. Indeed, prominences very often extend high into the corona. The larger the filament/prominence, the more likely it extends into the corona for several Mm up to several 10 Mm. For the filament undergoing reconnection in the present event, there are no simultaneous STEREO observations close to the limb. Therefore, the height of the filament and of the interacting flux on the other side of the current sheet cannot be inferred unambiguously. We can only resort to the extrapolated NLFFF and to our MHD model. From [44] we know that the filament is twisted. The NLFFF indeed shows twisted flux at the position of the filament on the west side of the current sheet (new Fig. 1b). This flux extends in a height range of roughly 5-15 Mm (new Suppl. Fig. 1). This same flux rises slowly under the photospheric driving in the MHD simulation. The resulting height range for the reconnection is $\sim 5-15$ Mm as well. This height range is clearly in the corona.

It is reasonable to expect that the H-alpha and EUV images show the same filament. This is the typical situation, with the EUV filament typically being somewhat larger and H-alpha showing only the coolest part of the filament (e.g., Heinzel, Schmieder, & Tziotziou 2001). In the present event, the filament is seen at 304 and 171 Å at the same position as the H-alpha filament (new Fig. 2 and Suppl. Movie 2), with indications of left-handed twist at all three wavelengths.

The bright structures seen in both H-alpha and the EUV show two types of structures: heated plasma in the current sheet (incl. the cusps) in the corona and ribbons (QSL footprints) in the chromosphere. The ribbons are identical at all wavelengths and most clear in H-alpha and 1600 Å when the emissions from the current sheet have faded (after $\sim 08:25$ UT); see the new Fig. 2, bottom row. The current sheet and cusp regions are more clear in the EUV at coronal and flare temperatures (171, 304, and 131 Å in Fig. 2, rows 1-3). However, they also show up as bright plasma in H-alpha because

cool plasma from the inflow volume, mainly from the filament, is heated in the current sheet (Fig. 2, rows 1-3). The differences in the images of the current sheet and cusps between H-alpha and the EUV are discussed in the text and in further detail below in the response to the comments by Reviewer 3.

The data include one piece of direct evidence for the involvement of the filament in the reconnection process. This is displayed in the new Fig. 1, panels c-h. Here the first snapshot (08:09:21 UT) shows a thread at the front edge of the filament already slightly separated from the main body of the filament (red arrow). This thread is obviously cut in the current sheet in the next frame (only the northern part can still be seen) and enters the reconnection outflow in the final four frames (blue arrows).

Additional support for the involvement of the filament in the reconnection process comes from the thread-like morphology of the bright H-alpha structures in the current sheet and cusps (see the brief discussion of these structures in the Results section, para 4 on pp. 4-5 and further detail in the response to the comments by Reviewer 3). This suggests an origin in reconnected and heated threads of the filament.

This comment and similar comments by the other reviewers have helped clarifying that the situation is different for the fibrils on the other side of the current sheet. These are typical chromospheric structures, and also our NLFFF shows them to be very low: in the range of 0-1 Mm. Therefore, it is flux extending upward from the fibrils into the corona, a system of magnetic loops shown in Figs. 1b and 8a-d, that reconnects at this side of the current sheet. Since the inflow into the current sheet is slowly driven from the photosphere, the fibrils at intermediate heights provide a valid estimate of the magnitude of the inflow. This underlying assumption of our analysis is now explicitly stated in the text on page 6, bottom para.

6) Fig 8 is not easy to understand. What is the height of plasmoids? Please enlarge the part of the plasmoids, and show magnetic field configuration around the plasmoids, especially 3D connectivity of each field line, much more clearly.

Reply: The updated Fig. 8 (now Fig. 9) shows the plasmoids in much greater detail. In particular, their 3D nature (small flux ropes) is much more clear. These flux ropes extend in a height range of about 5-15 Mm (Fig. 9b shows the slices through the middle of flux ropes), which is similar to the height extent of the current sheet between the reconnecting flux systems (see the updated Fig. 7 (now Fig. 8)). This is consistent with the fact that the plasmoids connect at their ends to the reconnecting and here mostly horizontal flux during most of their evolution. The axis of the small flux ropes follows the guide field component in the current sheet, which is roughly vertical in the considered event. The description of the

MHD simulation results is now extended to clearly present the properties of the plasmoids/small flux ropes (final para of this sub-section on pages 10 - 11).

7) In p8, the author wrote $B \sim 500\text{G}$. Is this true? It seems a bit strong for the average field strength around the coronal current sheet. If correct, would you show us evidence based on MHD modeling?

Reply: We have determined the field strength in a cross section of the flux rope shown in Fig. 1b (our NLFFF) at the point of closest contact with the current sheet (Suppl. Fig. 1). In a roughly circular area of $\sim 10\text{ Mm}$ diameter, the field strength exceeds 600 G, and the peak field strength in the center of the flux rope is $\sim 700\text{ G}$. For a conservative estimate of the released energy, we have used the value of $B \sim 500\text{ G}$ (see the text following Eq. 1).

Reviewer 2

1) The authors present picture book images of a solar flare. The images are well presented and the text is well written. The high-quality H alpha pictures seem to show a Petschek type reconnection. The images show four flows along magnetic separatrices. The measured velocities match the expectations.

To good to be true? Some doubts creep in when becoming aware that flares release most energy in non-thermal particles which heat the flare plasma to millions of degrees. Yet the authors put most weight on H alpha observations, tracing cool plasma. The question needs to be addressed: How relevant is H alpha? H alpha and EUV should be compared.

Reply: Brightenings of plasma during flares are often co-spatial and approximately synchronous from H-alpha to “warm/coronal” (few MK) EUV lines, and often up to the “hot/flare” EUV lines at $\sim 10\text{ MK}$. This is true also for the present flare; see the new Fig. 2 and the corresponding Suppl. Movie 2. Our response above to Comment 5 by Reviewer 1 argues that the filament is imaged by both H-alpha and EUV (304 & 171 Å), and the new Fig. 1 shows unambiguously that the filament is involved in the reconnection process (also see the response above). The comparison of the H-alpha and EUV images of the current sheet, cusps, and flare ribbons is done below in response to the detailed comment by Reviewer 3. All of these considerations are also included in the revised manuscript (see Results; High-res. Imaging on pages 3 - 6).

Recent work has further improved our knowledge about the partition of the released energy in flares. It was found that the ratio of non-thermal energy release (energetic particles) to thermal energy release (heating) decreases with decreasing flare magnitude. For low M-class flares, Warmuth & Mann (2020) find these components to be comparable (their Figs. 10 - 11) and the non-thermal energy release to be clearly lower than the bolometric

radiated energy (their Fig. 12). Both findings imply that the plasma at coronal to flare temperatures imaged by the EUV data (new Fig. 2) is directly heated by the flare energy release. The H-alpha data of higher resolution and clarity (less image saturation) map the same critical structures and allow a much more complete analysis of the reconnection flows; therefore, they are emphasized in the manuscript. The non-thermal energy release is considered as well to address possible implications for the mode of reconnection (Sub-section "X-ray spectroscopy of accelerated particles"), but is not a main focus in its own right in the given context.

2) The second question concerns the tracing of the magnetic field. Fibrils do not always trace the magnetic field (e.g. Asensio Ramos+ 2017). Figure 7 may suggest the answer, but needs to be discussed.

Reply: Based on the height estimate for the fibrils, 0-1 Mm in the NLFFF (see response to Comment 5 by Reviewer 1), we have changed the interpretation such that it is not the field in the fibrils but the field extending from the fibrils upward into the corona that reconnects in the event (Results section, para 2 on page 4). Under this interpretation, any small to moderate difference in the direction of the fibrils and the field, especially the relatively small spread of the field directions about the fibril orientation found by Asensio Ramos et al. 2017 and others, is not critical to our interpretation that the flare event provides images of the reconnection flows.

3) The blue-shift claimed in Fig. 3 seems to be marginal. Some more evidence needs to be presented.

Reply: Fig. 3 presents the Doppler measurement in the whole field of view of the EIS observation. Further spectroscopic observations, e.g., by IRIS, are not available. The blue shift in Fig. 3f is weak, but the highest velocities obtained from the spectral fits (40 km/s) are clearly above the noise level, which it is estimated to be 3 km/s (Kosugi et al. 2007). Moreover, the contiguous structure of the blue shifted region, especially in Fig. 4f, supports the interpretation that the upflows are real (each pixel in Fig. 4 represents an independent fitting of an individual EIS spectrum).

Furthermore, we expect the north edge of the current sheet to be higher than the south edge (i.e., an upward/downward component of the reconnection outflows) from the following independent facts which are all consistent with this assumption: (a) the NLFFF extrapolation and MHD modelling suggest that the current sheet is higher at the north end (new Fig. 8b-d); (b) the hard X-rays are emitted mainly from the south cusp, suggesting higher densities (a smaller height) in this location; (c) the projected reconnection outflows derived from the moving fibrils in Fig. 1,

are higher in the north region, suggesting a higher local Alfvén speed or motion away from the photosphere or both.

4) Finally, the reader would like not only to be presented a well-fitting case, but to know which of the many other observations of this flare remain unexplained.

Reply: Since this comment leaves open “which of the many other observations of this flare” the reviewer considers to be relevant, it is impossible to be sure that we have addressed the intended one. We have tried our best. We are aware that the active region had a very complex structure and that many other activities occurred before and after the period of reconnection analyzed here. The complexity of the magnetogram and its evolution, and their relevance for the triggering and termination of the reconnection event, were already addressed in the Discussion section, para 1 and Fig. 10 (which are not updated, except for very minor extensions in the interest of clarity). The possible relevance of the eruption in the neighbouring filament channel to the east for the triggering and termination of the reconnection event was also already addressed in the Discussion section, para 3 (now slightly shortened). Additionally, we have now pointed out and discussed the partial and confined eruption of the reconnecting filament during the impulsive phase of the considered flare (Results section, para 6-7 on pages 5-6). This process is a further potential effect that can lead to the termination of the reconnection in the event (Discussion section, new para 2).

Reviewer 3

1) The NVST movie is beautiful but unfortunately the Figure 1 does not give the same impression. It is very difficult to visualize with these snapshots what is important. In the Figure 1 the filament on the right and the chromospheric fibrils (whirls around the sunspot). These structures should be indicated in the Figure. In the movie we observe well these structures. They are very stable and not perturbed at all by the reconnection. Therefore it is difficult to agree with the sentence of the abstract: « The flare resulted from the interaction of a twisted filament and chromospheric fibrils ». In H α we probably see the bright flare ribbons at the QSL footprints in the low atmosphere, the filament being unchanged. The ribbons are blocked by the flux rope of the filament on one side and by the whirls on the other side.

Reply: Figure 1 was strongly updated to present the H α images in the same way (with the same intensity-color mapping) as in the NVST movie (Suppl. Movie 1) and to label all relevant structures as suggested. Other frames were selected for panels c-h, to unambiguously show the involvement of the filament in the reconnection process. In the original version of the figure, the brightness of the H α images was slightly saturated, to

show the similarity of the current sheet and cusps to the EUV images in several passbands (e.g., 304 and 171 Å). Four of the original H-alpha frames from Fig. 1c - h are now included in the new Fig. 2, and are displayed with the same color table as the H-alpha frames in Fig. 1 and the NVST movie.

Since the fibrils are very low in the atmosphere, we have changed the interpretation such that the fibrils only trace the bottom part of the flux entering the current sheet from the east side, i.e., remain below the height range of the reconnection in the corona (also see the responses to Comment 5 by Reviewer 1 and to Comment 2 by Reviewer 2). We have changed the sentence in the abstract to “The flare resulted from the interaction of a twisted magnetic flux rope surrounding a filament and nearby magnetic loops whose feet are anchored in chromospheric fibrils.” The title of the paper was updated accordingly.

On the other hand, we continue to be convinced that the filament seen in H-alpha was involved in the reconnection. The new Fig. 1c - h displays an episode of a filament thread slightly ahead of the main body of the filament being cut and its northern half entering the reconnection outflow, which unambiguously demonstrates the involvement.

We are convinced that the present event displays the usual relationship between H-alpha and EUV observations of a filament: both show the filament, with the “EUV filament” being larger because the “H-alpha filament” only shows the coolest parts of the structure (e.g., Heinzel et al. 2001). Both also indicate the twist of the magnetic flux rope the filament is embedded in. See detail below in the response to Comment (2).

Both H-alpha and the EUV/UV channels show the current sheet and cusps in the corona as well as the flare ribbons in the chromosphere. The ribbons are identical at all wavelengths and most clear in H-alpha and 1600 Å when the emissions from the current sheet have faded (after ~08:25 UT; see Fig. 2, bottom row). The current sheet and cusp regions are more clear in the EUV at coronal and flare temperatures (Fig. 2, rows 1 - 3). However, they also show up as emitting plasma in H-alpha because cool plasma from the inflow volume - partly from the filament, partly from the presumably “standard” corona on the other side of the current sheet - is heated in the current sheet. The new Fig. 2 and the corresponding new discussion of the images in the figure (Results section, para 4 - 5 on pages 4 - 5) explicitly address these relationships.

Since the H-alpha images have much higher resolution and are not saturated in intensity, they reveal substantial detail in the current sheet and cusp. During ~08:13 - 08:20 UT there is a prominent decrease of the H-alpha absorption in the whole area of the current sheet and cusp (Suppl. Movie 1). The central linear structure continuously seen during 08:05 - 08:10 UT (Fig. 1c - d, Fig. 2f, and Suppl. Movie 1) is co-spatial with the nearly vertical current sheet and the southwest separatrix extending from

the current sheet in the MHD modelling (updated Fig. 7 (now Fig. 8)). Subsequently appearing bright H-alpha structures are mostly thread-like and rapidly changing in structure throughout the flare (up to 08:24:17 UT); these are more suggestive of heated filament threads in the reconnection volume than of ribbons at the QSL footprints, due to their highly variable structure, even from frame to frame. This is addressed in the discussion of Fig. 2 as well.

It is true that many of the bright H-alpha structures in the current sheet and cusp region do not lie at the surface of the filament, but a considerable fraction does (see Suppl. Movie 1). Similarly, a dissolution of filament threads into the current sheet region, accompanied by brightening, is not the typical appearance, and is the most likely interpretation only between ~08:09:21 and ~08:10:58 UT (shown in Fig. 1) and between ~08:16:12 and ~08:20:14 UT (when a new thread forms in the southward reconnection outflow). A definitive interpretation of all the detailed features in the current sheet and cusp regions is not possible in view of their richness and complexity. We stress, however, that the H-alpha filament is involved in the reconnection and that the details of the features in the current sheet are not critical for our conclusions; we derive the global property of fast reconnection from the flows traced by the H-alpha structures in the inflow and outflow regions.

The very bright H-alpha structures in the four end regions of the separatrices, which are especially clear during the impulsive and main flare phases from 08:17 UT and most clear after ~08:25 UT, coincide with the footprints of the QSL in the new Fig. 8, panel e and with the flare ribbons in the 1600 Å passband. We agree that these H-alpha structures must be flare ribbons in the chromosphere (see the discussion of Fig. 2 in the para on pages 4 - 5).

2) Looking at the same time at the movies of AIA in the different wavelengths, we have a different interpretation of this reconnection and flare than what is proposed in the paper. In fact we visualize two arcades which are reconnected in the corona with two opposite cusps. The footpoints of these arcades are anchored in the QSLs between the filament and the chromospheric fibrils. The interaction with the filament and the whirls are not really visible. The movies of AIA are not shown in a Figure and not discussed in the paper. The last reconnected 131 Å and 94 Å loops are different from what is seen in H-alpha at 08:29 UT in Figure 1.

Reply: We had originally removed an overview figure of the EUV observations by SDO/AIA before the submission because of a similarity of the structures with those in the original Fig. 5a - f (now Fig. 6a - f) and kept only the corresponding Supplementary Movie. Now we have included representative EUV/UV images in (the new) Fig. 2 and discussed them in detail.

We agree with the referee's interpretation that the reconnecting flux at the east side of the current sheet has the structure of magnetic loops, similar to a loop arcade. This is indicated by the extrapolated NLFFF (updated Fig. 1b). However, the coronal field at the west side of the current sheet, which holds the filament, is more likely to be a twisted flux rope rather than a coronal arcade. The structure of a flux rope for the filament was inferred in [44] from observational data and is also indicated by some of the H-alpha frames in Suppl. Movie 1 (e.g. 08:09:21 UT, 08:15:48 UT), some of the 171 and 304 Å frames in Suppl. Movie 2 (e.g. 07:58:07 UT), and by our NLFFF (updated Fig. 1b). The twist parameter (Berger & Prior 2006) in the filament flux rope in the NLFFF reaches -2.6 (left-handed twist). It may also be noted that the specific structure of the plasma in the inflow region (flux rope vs. arcade) is not relevant for our interpretation of the reconnection event as fast, plasmoid-mediated reconnection.

Since the chromospheric fibrils ($z < 2$ Mm) are much lower than the filament (estimated height $z \sim 5 - 15$ Mm), they do not interact directly. We are grateful for pointing this out clearly and have corrected the description of the observations accordingly (see Results; High-res. Imaging), as well as the relevant sentence in the Abstract and the title of the paper. The chromospheric fibrils nevertheless trace the reconnection inflow, because the reconnecting loops at the east side of the current sheet are rooted in the area of the fibrils.

Using the comparison of H-alpha and EUV images in Fig. 2 and the comparison with the QSLs in the strongly updated Fig. 7 (now Fig. 8), we have also attempted to disentangle which bright structures in the images show the current sheet and cusps in the corona and which bright structures show the flare ribbons in the chromosphere. See the response to Comment (1) above.

The differences in the appearance of the current sheet and cusp region between H-alpha and the EUV are related to the different saturation of the images. The H-alpha images do not show saturation in extended areas. The EUV images are saturated in the current sheet and cusp area during the whole reconnection event analyzed, and even suffer from detector bleeding during the peak phase of the flare after 08:17 UT. If the H-alpha images are artificially saturated, as done in the original Fig. 1, their appearance is significantly closer to the 171 and 304 Å images in Fig. 2. For consistency of the presentation of the H-alpha data in Figs. 1 and 2 and in Suppl. Movie 1, and following the critique in Comment (1), no artificial saturation is applied to the H-alpha images in the revised manuscript.

We agree that an additional arc of hot plasma forms in the AIA 131 Å channel (~ 10 MK) at about the peak time of the flare. The sequence of AIA 304, 171, and 131 Å images in Fig. 2 shows that this arc results from the

confined partial eruption of flux out of the filament channel and involving part of the filament. The eruption starts approximately with the impulsive phase of the flare, with the precise time difficult to determine due to the strong detector saturation of the EUV images. It is possible that the retraction of the filament from the current sheet, beginning $\sim 08:14$ UT (Fig. 3c), is already related to the onset of this eruption. The Supplementary Movie 1 shows that the erupted part of the filament (and its enveloping flux rope) finds the final position around 08:32 UT and is then similar in shape to the hot arc seen in the 131 Å bandpass. Both the southern footpoint of the arc seen in Suppl. Movie 2 and the northern footpoint (outside the FoV of the animation) coincide with the filament footprints; they lie at $(x, y) \sim (-120, -230)$ and $\sim (-175, -70)$. The draining of the halted filament toward its southern footpoint is clearly seen in 171 and 304 Å and in H-alpha. By the end of the Supplementary Movie 1 (08:39 UT), the brightness of the flare has decreased sufficiently, so that both the erupted and remaining parts of the filament are visible. In the H-alpha movie, the partial eruption of the filament is not clear, but the draining of the erupted part toward the southern footpoint is clear, obviously due to sufficient cooling during the draining. Here again, it is clear that H-alpha and the EUV (304 and 171 Å) show one and the same filament and provide consistent information about its evolution. The partial eruption of the filament is a process separate from the reconnection that we study in our manuscript; moreover, it starts only shortly before or during the impulsive flare phase, while we analyze the reconnection before the onset of the impulsive phase.

3) In Figure 7 it is difficult to see exactly where should be the footprints of these reconnected arcades. May be we see two triangles at $z=0$ and they join at $z=2.9$ Mm. In d we see well the south arcade with the iso-surface. In e we see the arcade around the filament but it does not bring information. Vertical cuts should be done to show the two cusps.

Reply: Fig. 7 has been updated (newly done) as Fig. 8 and supplemented by an animation (Suppl. Movie 4) that shows horizontal cuts through the QSL structure throughout the relevant volume. This comprehensive new analysis of the QSLs, including the reconnecting current sheet, is extracted from the MHD simulation. The QSLs are now directly compared with the observed current sheet and separatrices (boundaries of the cusps) in H-alpha and the EUV, and nearly perfect agreement is obtained (Fig. 8e, f). The reconnected arcades in both reconnection outflow regions and their footprints (in QSL footprints) are seen in the 131 Å images in the new Fig. 2. We hope that the other points of critique of aspects in the old Fig. 7 are now obsolete.

REVIEWER COMMENTS

Reviewer #1 (Remarks to the Author):

The authors revised the text and figures significantly, considering the referees' comments. Especially the new figures 8 and 9 are excellent. Hence I now basically agree with publication of this paper in Nature communication. I have some minor questions and comments in the following. I will be happy if the authors consider these comments before publication.

1) In p 5, the authors wrote

“Additionally, the H alpha images show stationary brightenings, primarily as extensions of the cusps (contours in Fig. 2p), which are cospatial with traces of the main quasi-separatrix layers (QSLs) in the MHD model that extend from the current sheet and its separatrices down to the magnetogram plane (see Fig. 8 below). These brightenings are the classical H alpha flare ribbons in the chromosphere.”

However, above H alpha brightenings include the tip of the cusp at around (-240, -85). Is this also similar to the classical H alpha flare ribbons ? If so, where is other conjugate H alpha footpoint corresponding to this cusp ?

2) In Figure 3, it would be better if the time axis is also added at the top of the figure a and b.

3) In Supplementary Figures 1 and 2, the height is written in unit of arcsec. However, these heights should be written in unit of Mm.

Reviewer #2 (Remarks to the Author):

-

Reviewer #3 (Remarks to the Author):

I still like very much the NVST observations of a case study of magnetic reconnection.

The authors have taken into account the reviewer remarks and now it is clear that they analyzed only the preflare topology, the X point reconnection and the plasmoid formation.

This phenomena is visible in all the temperatures.

The title could be « Case study of a Petscheck reconnection in the solar atmosphere and plasmoid formation. »

I am worried that they used the term data driven simulation in the title. It is really misleading when we know that they used only one vector magnetogram before the reconnection. HMI vector magnetogram cadence is 12 minutes, SOT is even more.

They use the data-driven simulation and the Figure with the QSLs is impressive but the simulation works because of the high resolution mesh. It is a simulation and not NLFF extrapolation where even the FR is not detected. Therefore the reconnection between flux rope and arcade is really their personal assumption. The simulation even does not lead/show to the reconnection phase, the eruption and the confined flare.

Therefore I would recommend to clarify this point in the title and in the abstract.

« We conjecture that ...»

Response to referees

We thank the referees for their positive comments and further suggestions. We added a new Figure as supplementary Figure 1 to show the twist structure of the filament seen from H-alpha observations. We have responded fully as follows, so we hope the paper is now acceptable for publication.

Reviewer #1 (Remarks to the Author):

I now basically agree with publication of this paper in Nature communication. I have some minor questions and comments in the following. I will be happy if the authors consider these comments before publication.

1) In p 5, the authors wrote

“Additionally, the H alpha images show stationary brightenings, primarily as extensions of the cusps (contours in Fig. 2p), which are cospatial with traces of the main quasi-separatrix layers (QSLs) in the MHD model that extend from the current sheet and its separatrices down to the magnetogram plane (see Fig. 8 below). These brightenings are the classical H alpha flare ribbons in the chromosphere.”

However, above H alpha brightenings include the tip of the cusp at around (-240, -85). Is this also similar to the classical H alpha flare ribbons ? If so, where is other conjugate H alpha footpoint corresponding to this cusp ?

Reply: It has been our intention, here and in other places in the manuscript, to express that both, coronal structures – reconnecting current sheet and cusps – and chromospheric structures – flare ribbons –, are visible in H-alpha as well as in the EUV/UV channels of SDO/AIA. That is, we did not intend to suggest that the cusps are flare-ribbon structures at chromospheric heights. Indeed, three of the four groups of essentially stationary and roughly linear H-alpha structures (contours in Fig. 2p) show a separation from the cusps. Only the stationary H-alpha structure along the NW separatrix (or quasi-separatrix) overlaps with the cusp. There is more overlap in the EUV/UV images, which bring up fainter structures. Such overlap results from the projection effect. One can see in Fig. 8e–f ($z=2.5$ Mm) that the separatrix/QSL extends downward to the bottom plane, where its trace coincides with the H-alpha contours and also connects the H-alpha contours and the cusps. These connections mostly possess somewhat lower values of the squashing factor ($\log Q \sim 2$, cyan color) than the traces in the areas of the H-alpha contours. However, the MHD simulation indicates that the reconnection proceeds in the height range $z \sim 5\text{--}15$ Mm (Fig. 9d and associated text). We have rephrased the text on page 5, hoping that the above is now expressed clearly and unambiguously.

2) In Figure 3, it would be better if the time axis is also added at the top of the figure a and b.

Reply: We agree and have made the suggested changes.

3) In Supplementary Figures 1 and 2, the height is written in unit of arcsec. However, these heights should be written in unit of Mm.

Reply: We agree and have made the suggested changes

Reviewer #3 (Remarks to the Author):

1) I still like very much the NVST observations of a case study of magnetic reconnection. The authors have taken into account the reviewer remarks and now it is clear that they analyzed only the preflare topology, the X point reconnection and the plasmoid formation. This phenomena is visible in all the temperatures.

The title could be « Case study of a Petscheck reconnection in the solar atmosphere and plasmoid formation. »

Reply: Thank you. We agree and have removed the phrase "data-driven" from the manuscript and changed the title to "Multi-scale Observation and MHD Modeling of Magnetic Reconnection in a Confined Solar Flare". The reason is that the MHD modeling is also "multi-scale"; it includes both the large-scale configuration of the active region and the small-scale dynamics, i.e., plasmoids. Since "plasmoid-mediated reconnection" has become a fixed term in the plasma physics literature for a mode of reconnection that is different from Petschek reconnection (e.g., it is intrinsically non-stationary and operates in a long, initially Sweet-Parker-like current sheet), we are inclined to use the term plasmoid-mediated reconnection in this manuscript.

2) I am worried that they used the term data driven simulation in the title. It is really misleading when we know that they used only one vector magnetogram before the reconnection. HMI vector magnetogram cadence is 12 minutes, SOT is even more.

They use the data-driven simulation and the Figure with the QSLs is impressive but the simulation works because of the high resolution mesh. It is a simulation and not NLFF extrapolation where even the FR is not detected. Therefore the reconnection between flux rope and arcade is really their personal assumption. The simulation even does not lead/show to the reconnection phase, the eruption and the confined flare. Therefore I would recommend to clarify this point in the title and in the abstract.

« We conjecture that ... »

Reply: Thanks for the comment and to avoid any misleading, we have completely removed "data-driven" from the text and used "data-constrained" instead. The latter is justified because the initial condition of the MHD simulation is fully constrained by the observation data, using a MHD relaxation approach with the SDO/HMI vector magnetogram just prior to the reconnection phase analyzed in the manuscript.

It is our opinion that our analysis establishes strong evidence for the existence of a flux rope at the position of the filament. We have applied the nonlinear force-free field extrapolation method to the HMI vector magnetograms. The Wiegmann optimization code, which has become a standard in solar coronal research, yields a flux rope Fig. 1b and Suppl. Figs. 2–3 show this for the HMI magnetogram at 07:36 UT, and we have verified that this code also yields a flux rope in the extrapolation of the magnetogram at 08:00 UT, even closer to the analyzed reconnection event. The CESE-MHD NLFFF code developed by one of the authors (Jiang) does not yield a flux rope for this filament. From earlier research, it is well known that reconstructing a flux rope is a difficult task for nlfff extrapolation codes: carefully designed tests, using a numerically

constructed, quite realistic active-region model field that included a flux rope, showed that only two of the applied four nlfff codes could partly reconstruct the flux rope from the photospheric magnetogram (Metcalf et al. 2008 SoPhy 247, 269, Fig. 7). A similar result was obtained in DeRosa et al. (2009 ApJ 696, 1780). Therefore, if an nlfff code finds a flux rope, this is a strong indication that one indeed exists, and the opposite outcome is not a strong indication that no flux rope exists. Second, the flux rope found in our NLFFF agrees nearly perfectly in position with the observed filament; this is a strong indication that this NLFFF is a quite realistic model of the coronal field. Third, the H-alpha images contain some direct indications that the filament threads twist about each other (see the new Suppl. Fig. 1 and the frames at 08:15:48, 08:21:27, 08:23:28 and 08:24:17 UT in Suppl. Movie 1). Finally, also the genesis of the filament was found to indicate a flux rope structure [ref. 45].

The MHD simulation does clearly show magnetic reconnection at the observed x-y position. The plasmoids in the three simulation snapshots in Fig. 9 show the dynamics of the reconnection: repeated plasmoid formation and motion along the current sheet. It is correct that the simulation does not show reconnection between arcade field (on the east/left) side of the current sheet and a flux rope (on the west/right side); this is expressed in the manuscript by the sentence (in Revision 1) “although the twist in the magnetic flux of the filament is not recovered by the extrapolation technique used to construct the initial condition” (page 10). However, the reconnection between the arcade flux and the twisted filament is clearly shown by the H-alpha observations (Fig. 1c—h). Therefore, we felt entitled to write the penultimate sentence of the Abstract. In view of the reviewer's concern, we have changed the sentence as follows: “The analysis suggests that the flare resulted from ...”

We restrict the analysis to the X-shaped reconnection event in the long slow-rise phase of the flare because the observational data are heavily saturated during the subsequent impulsive flare phase. This is expressed in the manuscript. The soft X-ray flux during the reconnection event reaches the level of a substantial flare: M1.2 (page 6, middle para). Therefore, we feel entitled to use the term “confined flare” for the reconnection event.

Additional change:

The Discussion section was made more concise by rewording the final two sentences of para 3 and deleting para 4, which was more tentative than the other parts of the Discussion.

REVIEWER COMMENTS

Reviewer #1 (Remarks to the Author):

As for the two points I pointed out,

2) In Figure 3, it would be better if the time axis is also added at the top of the figure a and b.

3) In Supplementary Figures 1 and 2, the height is written in unit of arcsec. However, these heights should be written in unit of Mm.

I confirmed that the authors revised the figures.

As for the first point in p5 about the relation between H alpha bright region and classical H alpha ribbon, the authors replied and added some text. However, I do not still understand the explanation exactly. Hence I explained my question in the attached pdf file. I would be happy if authors answer to my question.

Response to referees

We again thank Reviewer #1 for raising the questions about the three forms of emitting structures to unambiguously clarify their nature and height:

- (1) the current sheet,
- (2) the attached two cusps, and
- (3) the four elongated structures (contours in Fig. 2p–t) that are arranged like extensions of the separatrices which branch off from the current sheet and enclose the cusps, especially in H α and 304 Å. For clarity, we enclose the PDF file that contains the questions and comments, and number them below.

REVIEWER COMMENTS

Reviewer #1 (Remarks to the Author):

As for the two points I pointed out,

- 2) In Figure 3, it would be better if the time axis is also added at the top of the figure a and b.
- 3) In Supplementary Figures 1 and 2, the height is written in unit of arcsec. However, these heights should be written in unit of Mm.

I confirmed that the authors revised the figures.

Reply: Thank you for your suggestions. Yes, we have done.

As for the first point in p5 about the relation between H alpha bright region and classical H alpha ribbon, the authors replied and added some text. However, I do not still understand the explanation exactly. Hence I explained my question in the attached pdf file. I would be happy if authors answer to my question.

Classical H alpha chromospheric ribbon as a result of coronal reconnection ?

Q1: Why this part [the current sheet and cusps] become bright?

Q2: Classical H alpha chromospheric ribbon as a result of coronal reconnection?

Q3: Is this [the structure indicated by the long red arrow] a result of local (chromospheric) reconnection at low height as suggested by the MHD model? This is NOT the classical H alpha ribbon?

Reply:

Our interpretation of these structures is as follows.

1) The current sheet and cusps are formed at coronal heights, i.e., both the EUV/UV and the H α emissions originate at coronal heights. The bright structure indicated by the green arrow in updated Figure 2l is not a classical H α ribbon. It is caused by reconnection in the coronal part of the current sheet that heats the plasma in the current sheet and its attached cusps, which themselves become bright in EUV/UV and H α , and this reconnection also provides the energy required for the other emissions.

2) The four elongated structures are classical chromospheric flare ribbons, often termed “H α ribbons”, although they are typically observed in H α and the EUV/UV in very similar form, as in the present event. As such, the energy that feeds them originates in coronal reconnection, identical to the energy transport in a usual flare (“standard flare model”).

This interpretation is adopted throughout the original and revised manuscript versions. With the previous and current revisions, we have attempted to express the interpretation and its underlying rationale more clearly; this required us to be slightly less concise.

To support the interpretation, we have performed additional analysis of the magnetic connections between the three categories of structures in the MHD model. **The new Supplementary Figure 4 shows the result of this analysis. The revised manuscript expresses the interpretation in improved form on pp. 4–5 and 11 (marked in bold).** For clarity, the Structures (1)–(2) are marked with a green arrow in the **updated Fig. 2l**, and Structures (3) are marked with cyan arrows in the **updated Fig. 2q**.

Our interpretation is based on the following considerations.

1) The reconnection proceeds primarily at coronal heights because it is observed to involve the filament (Fig. 1c–h). Filaments are known to be structures of chromospheric temperature suspended in the coronal magnetic field, i.e., at coronal heights. This is supported by the computed NLFFF model of the coronal field and by the data-constrained MHD model of the reconnection process, both of which indicate a height range of $z \sim 5\text{--}15$ Mm – clearly in the corona; see Suppl. Figs. 2–3 for the filament and Figs. 8–9 for the strongly reconnecting current sheet.

The current sheet and attached separatrices (or QSLs) in the MHD model actually extend down to the bottom of the box, which is the base of the corona (Fig. 8b–f). However, the current sheet becomes progressively shorter and reconnects less efficiently towards lower heights. In particular, the current sheet has an aspect ratio of $\sim 100\text{--}150$ at $z > 10$ Mm, >40 at $z > 5$ Mm, and >10 at $z > 2$ Mm, i.e., the fastest, plasmoid-mediated reconnection occurs only in the height range $z \sim 5\text{--}15$ Mm. Laminar reconnection flows extend further down to $z \sim 1$ Mm with decreasing extent for decreasing z . There are no reconnection flows at $z < 1$ Mm, i.e., the model does not suggest any role for chromospheric reconnection.

Therefore, the energy release occurs primarily in the corona. It is known that the heating of the plasma is realized primarily in the slow-mode shocks that form in the separatrices between the inflows and outflows (Petschek 1964). In plasmoid-mediated reconnection, small slow-mode shocks are attached to the individual X-lines, and the heating proceeds in the same manner in the reconnection outflows from each X-line into the adjacent plasmoids (e.g., Fig. 3 in Ni et al. 2015 [51]). This implies that the current sheet and the cusps are expected to be heated in the first instance. They will become bright (in the corona) if they include enough material to reach the emission measure required. In the analyzed event, the filament threads feed the current sheet and cusps with the necessary material. This is similar to the brightenings that are very often seen in the EUV/UV and $H\alpha$ at the surface of erupting filaments/prominences. Such brightenings are thought to result from reconnection with overlying flux, or when barbs tear off below. In summary, reconnection in the coronal part of the current sheet heats the plasma in the current sheet and its attached cusps, which themselves become bright in EUV/UV and $H\alpha$, and this reconnection also provides the energy required for the other emissions.

2) Energy transport in the corona is exclusively along the field. This applies to both relevant forms of energy transport: propagation of accelerated particles and heat conduction; both are negligible across the field in the corona. There are no vertical field lines from the reconnecting current sheet in the corona to the projected locations in the chromosphere directly underneath. The field threading the current sheet and its vicinity is much closer to the horizontal direction than to the vertical direction (Fig. 8b–d). The energy transport from the coronal energy release region to the chromosphere along these field lines does not lead to the traces of the separatrices (or QSLs) directly underneath the current sheet and cusps, but rather to the more remote extensions of the intersection of the separatrices (or QSLs) with the bottom plane of the MHD simulation. The bottom plane of the MHD simulation can be interpreted as the top layer of the chromosphere; this is the height where flare ribbons form. Therefore, and by their typical, essentially linear and somewhat irregular appearance, we interpret the more remote brightenings inside the contours in Fig. 2p–t as classical flare ribbons.

This is substantiated by the field line plot in the new Supplementary Figure 4. Here the field lines from the current sheet in the height range $z \sim 3\text{--}15$ Mm (cyan/green to red) are seen to have footpoints in the four elongated structures (3) which we interpret as classical chromospheric flare ribbons.

To summarise, we answer the questions as follows.

Q1: This region becomes bright because this is the energy release region and contains enough cool material (originating from the filament) that can be heated to turn $H\alpha$ and He II (304 \AA) into emission.

Q2: Yes.

Q3: (a) If the long red arrow points to the current sheet and cusp at coronal heights, then the answer is: No, the current sheet and cusps become bright as a result of reconnection in the corona (as suggested by the MHD model), and we agree that these are not classical $H\alpha$ ribbons.

(b) If the long red arrow points to the intersection of the separatrices (or QSLs) with the bottom of the box directly under the observed current sheet and cusps, then the answer is: No, there is no reconnection in the chromosphere and no emission from the chromosphere directly under the coronal current sheet and cusps.

(c) If the long red arrow points to the intersection of the separatrices (or QSLs) with the bottom of the box at the more remote locations of the elongated structures (3), then the answer is: No, this is the result of coronal reconnection in the observed current sheet, and these are indeed classical H α ribbons.

Additional changes:

1. We have chosen to change the emphasis in the **final sentence of the Abstract** from “hot flare plasma to cooler chromospheric structures” to “plasmoids and associated unresolved turbulent motions” because the 'cooler chromospheric structures' were found to be classical flare ribbons, while the occurrence of the plasmoid instability and turbulence in astrophysical reconnection events could be verified only rarely so far.

2. We have added one item of context information that distinguishes 3D reconnection (observed and modeled here) from the standard 2D picture. In 3D reconnection, the structure of a separator or quasi-separator (hyperbolic flux tube) is embedded, respectively, in the separatrices or quasi-separatrices (QSLs) that branch off from the current sheet at the tip of each cusp. This is added in the **Abstract (penultimate sentence)** and, similarly, in the **sentence that spans pages 5 and 6**.

Reviewer #1 (Remarks to the Author):

2021 Oct 3

Sorry for a delay of sending you my response.

I have been extremely busy in the last week, but now have a time to carefully check the authors' revision and replies, and confirmed that all my questions have been answered and solved enough. Hence I now recommend publication of this paper. This is an excellent paper.